# Flexible power generators by Ag₂Se thin films with record-high thermoelectric performance

Dong Yang[1,2,5], Xiao-Lei Shi[3,5], Meng Li[3], Mohammad Nisar[1], Adil Mansoor[1], Shuo Chen[1], Yuexing Chen[1], Fu Li[1], Hongli Ma[2], Guang Xing Liang[1], Xianghua Zhang[2], Weidi Liu[3,4], Ping Fan[1], Zhuanghao Zheng[1]✉ & Zhi-Gang Chen[3]✉

Exploring new near-room-temperature thermoelectric materials is significant for replacing current high-cost $Bi_2Te_3$. This study highlights the potential of $Ag_2Se$ for wearable thermoelectric electronics, addressing the trade-off between performance and flexibility. A record-high $ZT$ of 1.27 at 363 K is achieved in $Ag_2Se$-based thin films with 3.2 at.% Te doping on Se sites, realized by a new concept of doping-induced orientation engineering. We reveal that Te-doping enhances film uniformity and (00$l$)-orientation and in turn carrier mobility by reducing the (00$l$) formation energy, confirmed by solid computational and experimental evidence. The doping simultaneously widens the bandgap, resulting in improved Seebeck coefficients and high power factors, and introduces $Te_{Se}$ point defects to effectively reduce the lattice thermal conductivity. A protective organic-polymer-based composite layer enhances film flexibility, and a rationally designed flexible thermoelectric device achieves an output power density of 1.5 mW cm⁻² for wearable power generation under a 20 K temperature difference.

With the increasing popularity of smart wearable electronic devices such as wireless headphones, smart glasses, and smartwatches, traditional battery power sources have become inadequate to meet the demand for long-term use[1]. There is an urgent need in the market to develop sustainable power supply technology. Among the various emerging power generation methods, flexible thermoelectric devices (F-TEDs) that can directly convert temperature differences between the human body and the external environment into electricity, have gained significant attention[1]. F-TEDs are easy to integrate into wearables, have no mechanical vibrations, and can provide sustainable power without the need for maintenance, making them an ideal self-

powered technology[2]. F-TEDs are composed of pairs of p- and n-type thermoelectric materials connected in electrical series and thermal parallel[3]. To improve the energy conversion efficiency of F-TEDs, both p- and n-type materials require high thermoelectric performance[2]. This thermoelectric performance can be determined by the figure-of-merit $ZT = S^2\sigma T/\kappa = S^2\sigma T/(\kappa_e + \kappa_l)$, in which $S^2\sigma$ is the power factor of the material, determined by the Seebeck coefficient $S$ and electrical conductivity $\sigma$, $\kappa$ represents the total thermal conductivity of the material, determined by electronic thermal conductivity $\kappa_e$ and lattice thermal conductivity $\kappa_l$, and $T$ represents the absolute temperature[4]. Generally, $S$, $\sigma$, and $\kappa_e$ are strongly correlated with the carrier concentration $n$ of

[1]Shenzhen Key Laboratory of Advanced Thin Films and Applications, Key Laboratory of Optoelectronic Devices and Systems of Ministry of Education and Guangdong Province, College of Physics and Optoelectronic Engineering, Shenzhen University, Shenzhen, Guangdong 518060, China. [2]Univ Rennes, CNRS, ISCR (Istitut des Sciences Chimiques de Rennes) UMR 6226, Rennes F-35000, France. [3]School of Chemistry and Physics, ARC Research Hub in Zero-emission Power Generation for Carbon Neutrality, and Centre for Materials Science, Queensland University of Technology, Brisbane, QLD 4001, Australia. [4]Australian Institute for Bioengineering and Nanotechnology, The University of Queensland, Brisbane, QLD 4072, Australia. [5]These authors contributed equally: Dong Yang, Xiao-Lei Shi. ✉e-mail: zhengzh@szu.edu.cn; zhigang.chen@qut.edu.au

the material, so optimizing $ZT$ can be achieved by adjusting $n$ through rational doping or alloying methods[5]. Oppositely, $\kappa_l$ is weakly correlated with $n$ but can be further reduced by introducing additional crystal and lattice defects to scatter heat-carrying phonons with different wavelengths. However, these defects may simultaneously scatter charge carriers, reducing the carrier mobility $\mu$ of the material and in turn, $\sigma$. Therefore, optimizing the performance of thermoelectric materials has always been a challenging research task.

As key components in F-TEDs, thermoelectric materials must simultaneously possess high performance and certain flexibility. Compared with inorganic materials, organic conducting polymers and organic/inorganic composites show relatively low $ZT$ values although they usually have higher flexibility due to the nature of molecular chains as matrix[3]. Therefore, current research mostly focuses on making high-performance inorganic materials flexible[6]. For example, the designs of flexible inorganic thin films, aim to enhance their flexibility while maintaining their high performance[6]. Since F-TEDs are typically used in wearable scenarios, the materials inside the devices need to have high $ZT$s near room temperatures. Thus, materials meeting this criterion are generally traditional $Bi_2Te_3$-based thermoelectric thin films, known for their high thermoelectric performance near room temperatures, such as Ag-doped $Bi_2Te_3$ highly oriented thin films reported to have a $ZT$ as high as 1.2[6]. However, $Bi_2Te_3$ has relatively poor flexibility due to its crystal structure, and the low natural abundance of tellurium (Te) makes it relatively expensive, leading to cost inefficiency. Therefore, the development of alternative inorganic thermoelectric thin films is of paramount importance.

N-type silver selenide ($Ag_2Se$) exhibits characteristics of an "electronic crystal and phonon liquid," with intrinsic high $\sigma$ and low $\kappa$ at near-room temperatures[7]. Additionally, it is abundant in Earth's crust, environmentally friendly, and represents an ideal alternative to $Bi_2Te_3$[8]. $Ag_2Se$ possesses a stable orthorhombic structure at lower temperatures, and is considered a typical narrow-bandgap semiconductor with bandgap widths ranging from 0.02 eV to 0.22 eV[7,8]. $Ag_2Se$ transforms from orthorhombic phase to cubic phase above 130 °C, and most research focuses on the near-room-temperature phase of $Ag_2Se$[8]. Besides, $Ag_2Se$ exhibits a certain ductility due to its crystal structure[3]. To enhance its practical utility, improvements in both thermoelectric performance and flexibility are required for $Ag_2Se$-based thin films. To date, many efforts have been made to achieve these goals, such as tuning the ratio of Ag and Se[9], doping with elements Cu[10], S[11], and Ga[12], alloying with other elements (e.g., Se[13] and Ag[14]) and compounds (e.g., $CuAgSe$[14]), and hybridizing with carbons (e.g., single-walled carbon nanotubes (SWCNTs)[15]) and conducting polymers (e.g., polyvinylpyrrolidone (PVP)[16], polypyrrole (PPy)[13], poly(3,4-ethylenedioxythiophene) (PEDOT)[17] and polyvinylidene fluoride (PVDF)[18]). Benefiting from these novel strategies, high $ZT$s of ~1 have been achieved. To enhance the flexibility of the thin film, flexible substrates or supporters are usually needed, such as polyimide (PI)[9,10,19,20], Nylon[12–17,21–25], scaffold[26], flexible glass[18,27–29], polyethylene terephthalate (PET)[11], and paper[30]. However, further enhancing its thermoelectric performance and flexibility remains an ongoing goal.

In addition to conventional doping- and alloying-based compositional optimizations, designing highly oriented polycrystalline thin films is one of the effective methods to improve their overall thermoelectric performance[6,31]. Both theoretical and experimental results showed that (00$l$)-textured film can lead to a higher $S$ than that of the (013)-textured $Ag_2Se$-based film, contributing to an improved $S^2\sigma$, which can be achieved by tuning the ratio of Ag and Se[20]. However, some other studies argue that films with the (00$l$)-preferred orientation exhibit considerably lower electrical transport performance compared to that with the (013)-preferred orientation[21]. This is mainly due to the fact that the $\sigma$ in the (00$l$) orientation does not reach the same high levels as in the (013) orientation[8]. Therefore, there is still some controversy regarding the adjustment of film orientation to enhance thermoelectric performance. Nevertheless, it is undeniable that optimizing $n$ and improving $\mu$, whether it be to enhance $\sigma$ or increase the $S$, remain focal points. As for the methods to achieve this orientation, there is currently no definitive consensus. One of the theories that can be considered is based on the Gibbs-Wulff crystal growth law[32], suggesting that the introduction of micro-strain due to doping and the substitution of Se positions may result in lower free energy on the (00$l$) plane, leading to faster growth rates. Consequently, the controlled concentration of element doping can lead to changes in overall orientation.

Based on this theory, in this work, we undertake theoretical calculations to confirm that Te-doping can reduce the formation energy of the (00$l$) plane, resulting in a strong (00$l$) orientation across the film to enhance its uniformity. Figure 1a, b show unit cells of $Ag_2Se$ before and after Te-doping viewed along the $c$- and $a$-axis, respectively. Based on the computational results by spin-polarized density functional theory (DFT), the (001) surface energies of pristine and Te-doped $Ag_2Se$ are calculated to be 0.004 and 0.002 eV/Å$^2$, respectively, indicating that the introduction of Te on Se sites is beneficial for the formation of the (001) surface of $Ag_2Se$. Other computational method based on the commercial software package Material Studio shows the same results (see Supplementary Fig. 1). Generally, the uniform thin-film structure usually results in high $\sigma$. To verify the as-calculated results, we fabricate $Ag_2Se$ thin films without and with Te doping by a vacuum thermal co-evaporation method, as illustrated by Fig. 1c, d, respectively. Here PI are used as flexible substrates to support the as-deposited thin films. The transmission electron microscopy (TEM) image in Fig. 1c indicates a typical polycrystalline feature of pristine $Ag_2Se$ thin film with anisotropic grains, while the TEM image in Fig. 1d exhibits a highly (00$l$)-orientated feature of Te-doped $Ag_2Se$ thin film. Figure 1e compares the $S^2\sigma$ and $ZT$ values between our Te-doped $Ag_2Se$ thin film and reported works[9,13,14,17,19,21–23,27,28,30,33–35], indicating that our as-achieved thermoelectric performance ranks as top values. Such high performance is derived from the Te-doping-induced suppression of $n$, which significantly improves the $S$. Also, Te-doping results in a strong (00$l$) orientation within the film to enhance its $\mu$, which helps keep a high $\sigma$ and in turn a high $S^2\sigma$. Additionally, Te-doping-induced $Te_{Se}$ point defects effectively scatter phonons, effectively reducing $\kappa$. To further improve the flexibility of the as-fabricated thin film, we designed a polymer-hybrid-based coating composed of 90% poly(vinyl laurate) and 10% N-methylpyrrolidone on the thin film surface. Figure 1f compares measured increased normalized resistance $\Delta R/R_0$ of Te-doped $Ag_2Se$ thin films with and without coating as a function of bending radius. The insets show the illustration of the spin coating process (left) and the photo of testing the flexibility of the as-fabricated thin film (right). After coating the as-designed polymer-hybrid-based protect layer, the flexibility is significantly improved. Finally, we assemble an F-TED using a pair of p-type $Sb_2Te_3$ and n-type $Ag_2Se$ films, realizing an open-circuit voltage of 6 mV, an output power of 65 nW, and a power density of 1.5 mW cm$^{-2}$ under a temperature difference $\Delta T$ of 20 K. The as-designed thin film and device demonstrate high potential for application in wearable power generation.

## Results and discussion

To determine the optimal Te doping concentration for achieving the best thermoelectric performance in $Ag_2Se$ thin films, we chose various ratios of elemental powders as raw materials, defining a series of nominal compositions $Ag_2SeTe_\delta$ ($\delta = 0$, 0.02, 0.04, 0.06, 0.08, 0.10, and 0.12), during the thin-film preparation process. Table 1 compares nominal and actual compositions of as-fabricated Te-doped $Ag_2Se$ thin films, in which the actual compositions of the thin films were obtained from the energy dispersive spectroscopy (EDS) results. It is evident that the actual Te contents in the films do not perfectly match the nominal compositions. This discrepancy arises because during the deposition process, a portion of the Te can be evacuated, and both Se

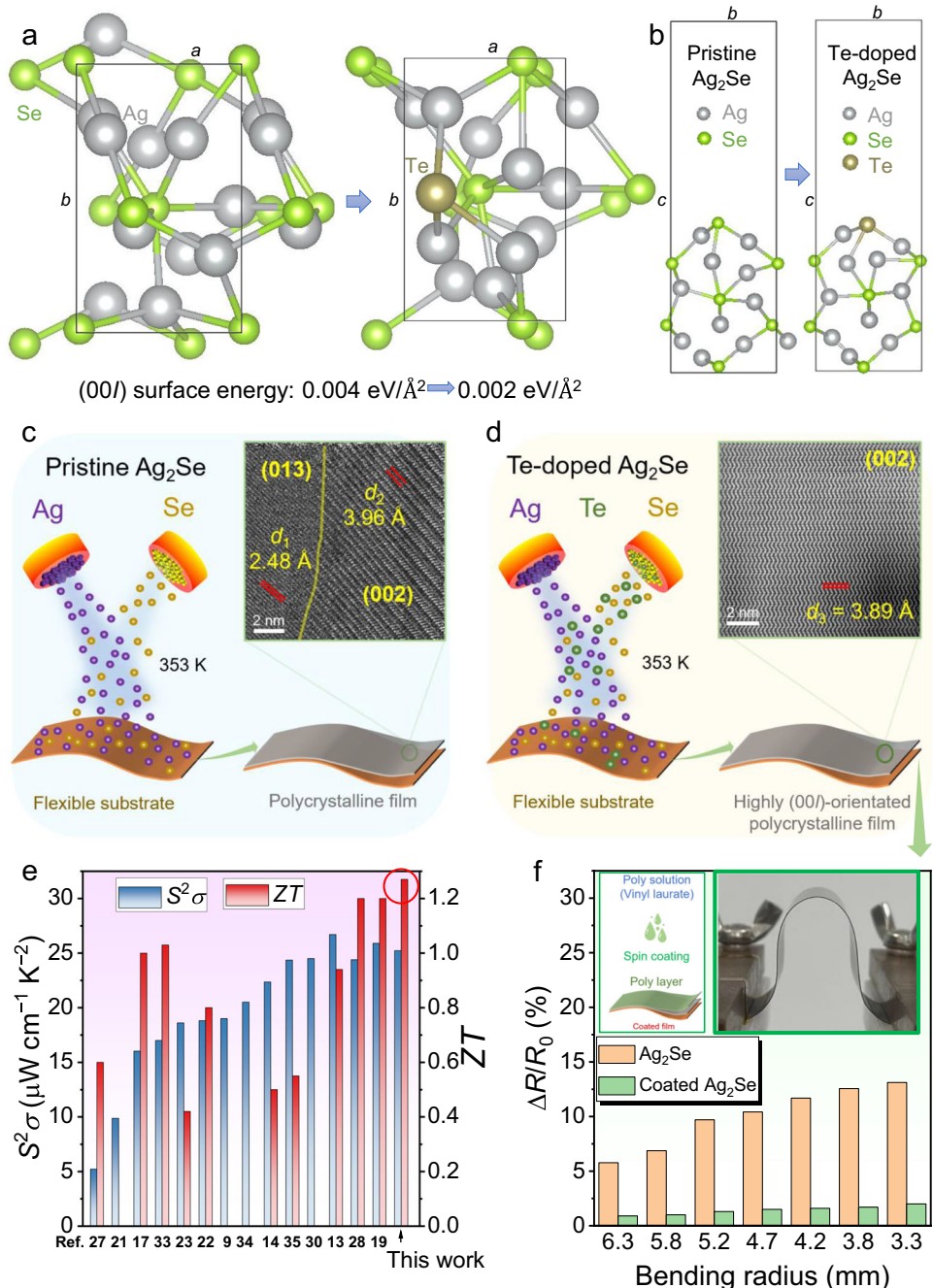

**Fig. 1 | Introduction of Ag₂Se thin films with both high thermoelectric performance and flexibility by Te-doping-induced band and orientation engineering.** Unit cells of Ag₂Se before and after Te-doping viewed along the (**a**) *c*- and (**b**) *a*-axis. Illustrations of fabricating Ag₂Se thin films (**c**) without and (**d**) with Te-doping by a vacuum thermal co-evaporation method. The transmission electron microscopy (TEM) image in **c** indicates the polycrystalline feature of pristine Ag₂Se thin film, and the TEM image in (**d**) indicates the highly (00*l*)-orientated feature of Te-doped Ag₂Se thin film. **e** Comparison of power factor $S^2\sigma$ and dimensionless figure-of-merit *ZT* values between this work and reported works including printed Ag-Se-based thin film[27], Ag₂Se film on nylon membrane[21], poly(3,4-ethylenedioxythiophene) (PEDOT)/Ag₂Se/CuAgSe composite film[17], printed β-Ag₂Se[33], Ag/Ag₂Se composite film[23], Ag₂Se film on porous nylon membrane[22], Ag₁.₈Se film[9], Ag₂.₀₆Se film[34], Ag₂Se/Ag/CuAgSe composite film[14], Ag/ Ag₂Se composite film[35], Ag-rich Ag₂Se film[30], Ag₂Se/Se/polypyrrole (PPy) composite film[13], Ag₂Se film prepared by pulsed hybrid reactive magnetron sputtering (PHRMS)[28], and microstructurally tailored β-Ag₂Se thin film[19] (from left to right). **f** Measured increased normalized resistance $\Delta R/R_0$ of Te-doped Ag₂Se thin films with and without coating as a function of bending radius. The coating is composed of 90% poly(vinyl laurate) and 10% N-methylpyrrolidone. The insets show the illustration of the spin coating process (left) and the photo of testing the flexibility of the as-fabricated thin film (right).

and Te can volatilize from the surface during the heat treatment process. Consequently, the final actual compositions differ somewhat from the nominal compositions. Fortunately, the actual Te contents in the films were also increased with increasing the nominal compositions of Te contents. This suggests that Te-doping can be successfully achieved, and no doping limit has been observed thus far. This indicates that under low-concentration Te-doping conditions, Te-doping has a high success rate. To better analyze the impact of Te doping on the thermoelectric performance of the films, we used the actual atomic concentration of Te as a variable, denoted as *x* (at.%) to assess the

**Table 1 | Comparison of nominal and actual compositions of as-fabricated Te-doped Ag₂Se thin films**

| Nominal composition | Ag (at%) | Se (at%) | Te (at%) |
|---|---|---|---|
| $Ag_2Se$ | 66.8 | 33.2 | 0 |
| $Ag_2SeTe_{0.02}$ | 66.4 | 32.9 | 0.7 |
| $Ag_2SeTe_{0.04}$ | 66.0 | 32.7 | 1.3 |
| $Ag_2SeTe_{0.06}$ | 65.6 | 32.5 | 1.9 |
| $Ag_2SeTe_{0.08}$ | 65.2 | 32.2 | 2.6 |
| $Ag_2SeTe_{0.10}$ | 64.7 | 32.1 | 3.2 |
| $Ag_2SeTe_{0.12}$ | 64.3 | 31.9 | 3.8 |

influence of different Te doping concentrations on thermoelectric properties.

To understand the phase information of as-fabricated Te-doped Ag₂Se thin films, X-ray diffraction (XRD) was investigated. Figure 2a compares XRD patterns of Ag₂Se thin films with different Te concentrations $x$ ($x$ = 0, 0.7, 1.3, 1.9, 2.6, 3.2, and 3.8 at.%) with $2\theta$ ranges from 20° to 55°. All diffraction peaks perfectly match the room-temperature phase of Ag₂Se, and there is no presence of impurity phases within the precision of the XRD examination. Noticeably, the pristine Ag₂Se thin film without Te-doping exhibits a typical poly-crystalline feature with anisotropic grains since many different diffraction peaks were observed. Oppositely, the Ag₂Se thin film with Te-

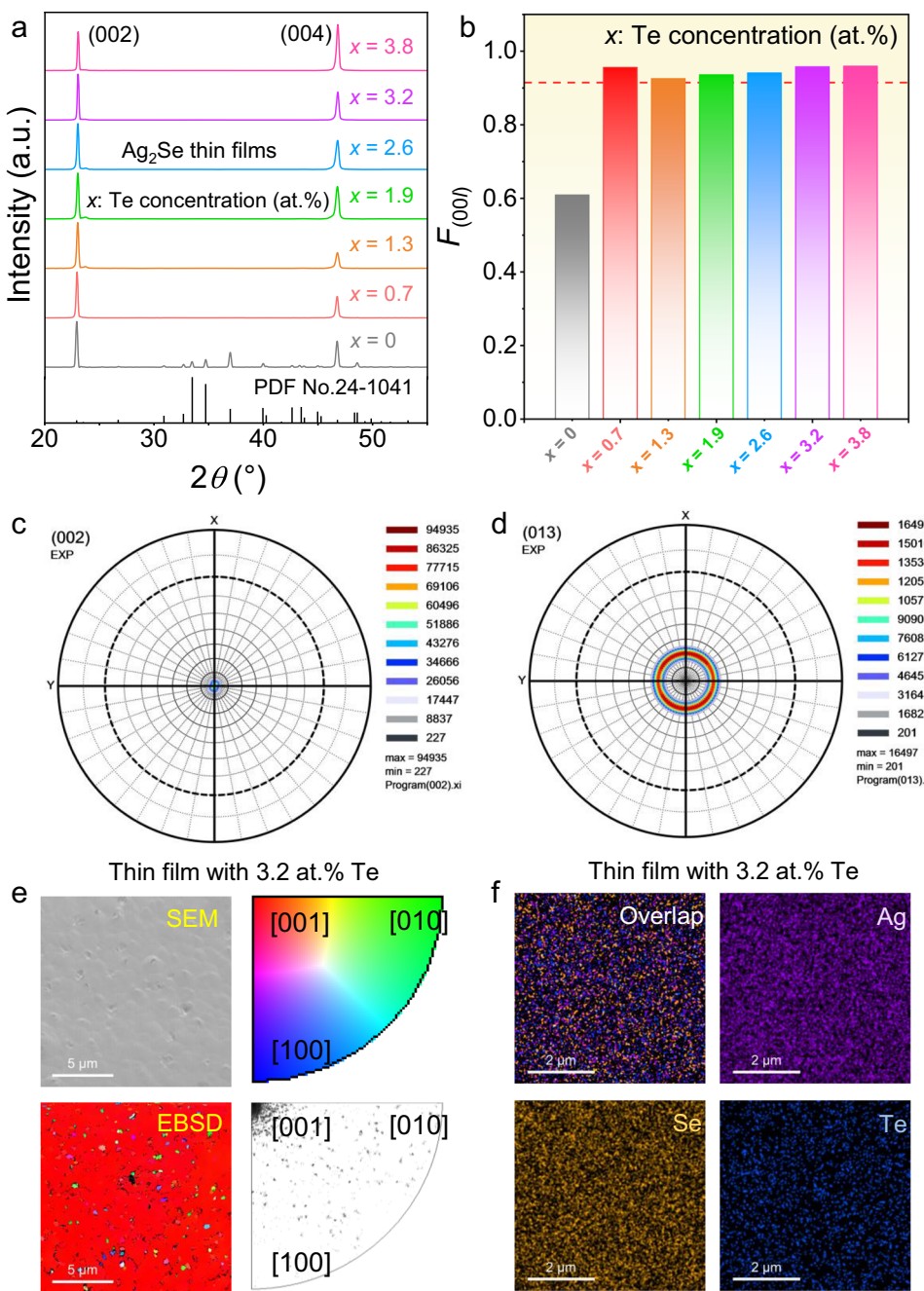

**Fig. 2 | Phase, structural, and compositional information of as-fabricated Te-doped Ag₂Se thin films. a** X-ray diffraction (XRD) patterns of Ag₂Se thin films with different Te concentrations $x$ ($x$ = 0, 0.7, 1.3, 1.9, 2.6, 3.2, and 3.8 at.%) with $2\theta$ ranges from 20° to 55°. **b** Determined orientation factors $F_{(00l)}$ of the films. Pole files of Ag₂Se thin films with 3.2 at.% Te along the (**c**) (002) and (**d**) (013) directions. **e** Scanning electron microscopy (SEM) and corresponding electron back-scattered diffraction (EBSD) images of Ag₂Se thin films with 3.2 at.% Te from the top view. The orientation information is provided for reference. **f** Energy dispersive spectroscopy (EDS) maps of Ag₂Se thin films with 3.2 at.% Te for overlap and individual Ag, Se, and Te elements.

doping exhibits a typical highly orientated feature with isotropic grains since only (00$l$) peaks ((002) and (004) peaks) were observed. Figure 2b compares determined orientation factors $F_{(00l)}$ of the films. The calculation methods of $F_{(00l)}$ can be referred to the experimental details. After Te-doping, the $F_{(00l)}$ values were significantly enhanced to >0.9, indicating improvements in the overall orientation of the thin films. Figure 2c, d compare the pole files of Ag$_2$Se thin films with 3.2 at.% Te along the (002) and (013) directions, respectively. The intensities belonging to the (00 $l$) become more significant, double-confirming the highly (00$l$)-oriented feature of the thin film.

To understand the structural and compositional information of as-fabricated Te-doped Ag$_2$Se thin films, scanning electron microscopy (SEM), electron back-scattered diffraction (EBSD), and EDS analysis were investigated. Figure 2e exhibits SEM and EBSD images of Ag$_2$Se thin films with 3.2 at.% Te from the top view. The orientation information is provided for reference. As can be seen, the as-fabricated thin film with Te-doping is relatively smooth and dense without significant pores, indicating a high thin-film quality. The EBSD results indicate that the as-fabricated thin film possesses significant [001] orientation, confirming the highly oriented feature. Figure 2f exhibits EDS maps of Ag$_2$Se thin films with 3.2 at.% Te for overlap and individual Ag, Se, and Te elements. All elements are uniformly distributed without obvious composition segregation, indicating that the doping is uniform at a microscale. The TEM results of pristine Ag$_2$Se thin film are provided in Supplementary Fig. 2, which shows a typical polycrystalline feature with anisotropic grains, and the SEM and corresponding EDS results of Ag$_2$Se thin films doped with different concentrations of Te (excluding 3.2 at.% Te) can be referred to Supplementary Figs. 3–7 for reference.

To understand the micro/nanostructural and compositional information of as-fabricated Te-doped Ag$_2$Se thin films, aberration-corrected scanning TEM (Cs-STEM) characterizations were investigated. Figure 3a shows a TEM high-angle annular dark field (HAADF) image of the Ag$_2$Se thin film with 3.2 at.% Te from a cross-sectional view. The sample was prepared by a focused ion beam (FIB) technique. The Te-doped Ag$_2$Se thin film exhibits a good quality with no apparent pore or noticeable impurity phase from a microscale. The thickness of the as-fabricated thin film is relatively uniform. In comparison to the undoped sample (Supplementary Fig. 2), their grain orientation is more uniform. Figure 3b shows EDS maps for overlap and individual Ag, Se, and Te elements, respectively, taken from Fig. 3a. All elements are uniformly distributed, and there is no noticeable element segregation. Figure 3c shows an EDS line scan taken from Fig. 3a. Along the thickness direction of the film, there is no variation in element distribution, further confirming that doping is exceptionally uniform. Figure 3d shows a Cs-STEM HAADF image taken from Fig. 3a. Even in the area close to the flexible substrate (PI), the as-prepared film maintains excellent crystallinity and orientation with no noticeable defects in this region. Furthermore, there is no apparent crack between the film and the substrate, indicating a strong adhesion between them. Figure 3e shows a magnified Cs-STEM HAADF image taken from Fig. 3d. The inset shows the corresponding fast Fourier transform (FFT) pattern with indexed information, indicating that the viewed direction is $[2\bar{1}0]$. The indexed high-resolution FFT pattern can also be referred to Supplementary Fig. 8. As can be seen, the microregion of the film exhibits excellent orientation and crystallinity. Figure 3f shows corresponding strain maps along different directions. Interestingly, the strain maps reveal a more pronounced strain along the $y$-direction, indicating that strains primarily affect the $y$-direction. This is likely attributed to Te substitutional doping. Figure 3g shows a high-resolution Cs-STEM HAADF image taken from Fig. 3e. The inset shows the crystal structure of Te-doped Ag$_2$Se for reference. Figure 3h shows a filtered Cs-STEM HAADF image,

showing the contrast difference. The arrow indicates a potential point defect of Te$_{Se}$ since it exhibits a higher contrast difference. Figure 3i also shows a line profile taken from Fig. 3g, confirming the potential point defect of Te$_{Se}$. The EDS maps taken from such a high magnification also confirm the potential point defect of Te$_{Se}$, and indicate that these point defects may be not uniformly distributed at a nanoscale (Supplementary Fig. 9) because the vacuum thermal co-evaporation method may not concisely dope Te on Se sites at an atomic level, which is understandable.

In terms of the effect of Te-doping on the thermoelectric performance of Ag$_2$Se thin films, we evaluated the thermoelectric properties of Ag$_2$Se thin films with different Te concentrations $x$ ($x$ = 0, 0.7, 1.3, 1.9, 2.6, 3.2, and 3.8 at.%). Figure 4a-c show temperature-dependent $\sigma$, $S$, and determined $S^2\sigma$. Generally, with increasing the $x$ value, $\sigma$ is decreased and $S$ is improved, leading to an optimized $S^2\sigma$ of 24.8 µW cm$^{-1}$ K$^{-2}$ at 363 K when 3.2 at.% Te was doped into the thin film. To understand the variations of $\sigma$, $S$, and $S^2\sigma$, Fig. 4d compares measured $x$-dependent $n$ and $\mu$ at room temperature. By increasing the $x$ value, the $n$ suppresses but the $\mu$ enhances. The improvement in $\mu$ benefits from the more uniform structures of the thin films with highly (00$l$)-orientations, which help maintain high $\sigma$ values even though $n$ is decreased. To explain the decreased $n$, we perform first-principles DFT calculations. Figure 5a, b compare calculated band structures of Ag$_2$Se (Ag$_{24}$Se$_{12}$) and Te-doped Ag$_2$Se (Ag$_{24}$Se$_{11}$Te$_1$), respectively, indicating that Ag$_2$Se is a typical narrow-gap semiconductor, despite doping with Te or not. Interestingly, after doping Te on Se sites, the bandgap was slightly increased from 0.061 eV to 0.078 eV, explaining the improved $S$ and reduced $n$. Figure 4e compares calculated $x$-dependent effective mass $m^*$ and deformation potential $E_{def}$ at room temperature by a single parabolic band (SPB) model. With increasing the $x$ value, the $m^*$ drops but the $E_{def}$ enhances. The decrease in $m^*$ comes from the suppression of $n$ since there is no obvious secondary phase to provide a phase boundary that can trigger the energy filtering effect[36]. This also confirms that the increased $S$ does not come from the energy filtering at interfaces. In terms of the $E_{def}$, its improvement indicates that the deformation capability of the lattice becomes more difficult due to the Te doping. We also compare calculated and experimental $S^2\sigma$ as a function of $n$ (Supplementary Fig. 10). The calculated data are from the SPB model. As observed, the $n$ values are higher than those reported for Ag$_2$Se single crystals[37], and they can be progressively optimized by increasing the Te doping concentration to achieve peak $S^2\sigma$ in the thin film. Figure 4f shows measured temperature-dependent $\kappa$ for Ag$_2$Se thin films with different Te concentrations $x$ ($x$ = 0, 0.7, 1.3, 1.9, 2.6, 3.2, and 3.8 at.%). With increasing the $x$ value, $\kappa$ is decreased, indicating that Te-doping can help suppress the overall thermal transport of the thin film. Figure 4g compares $x$-dependent electronic and lattice thermal conductivities ($\kappa_e$ and $\kappa_l$) at room temperature. The $\kappa_e$ data are from the formula $\kappa_e = L\sigma T^3$, where $L$ is the Lorenz number (Supplementary Fig. 11), and $\kappa_l = \kappa - \kappa_e$. Generally, with increasing the $x$ value, both the $\kappa_l$ and the $\kappa_e$ decrease. The decrease in $\kappa_e$ comes from the suppression of $\sigma$, while the dropped $\kappa_l$ is derived from the increased intensity of Te$_{Se}$ point defects that effectively scatter phonons. It should be noted that when the atomic content of Te is 3.2 at.%, the $\kappa_l$ obtained is quite low. This is partly due to the fact that thermal-conductivity measurements of thin films typically have a significant margin of error. Although methods for measuring the in-plane $\kappa$ of polymer thin films have been reported[38], the current technology for testing the in-plane $\kappa$ of inorganic thin film is not yet fully mature[39]. Additionally, calculations using the SPB model also exhibit some deviation, and this deviation varies depending on the material system[40]. Figure 4h shows determined temperature-dependent $ZT$. An outstanding $ZT$ value of 1.27 at 363 K, as well as a high $ZT$ of 1.15 at room temperature, both rank top values among reported ones to date (Supplementary Table 1). Figure 4i also compares calculated and experimental $ZT$ as a function of $n$. The calculated data are from the SPB model. As can be seen, the $n$ value can be

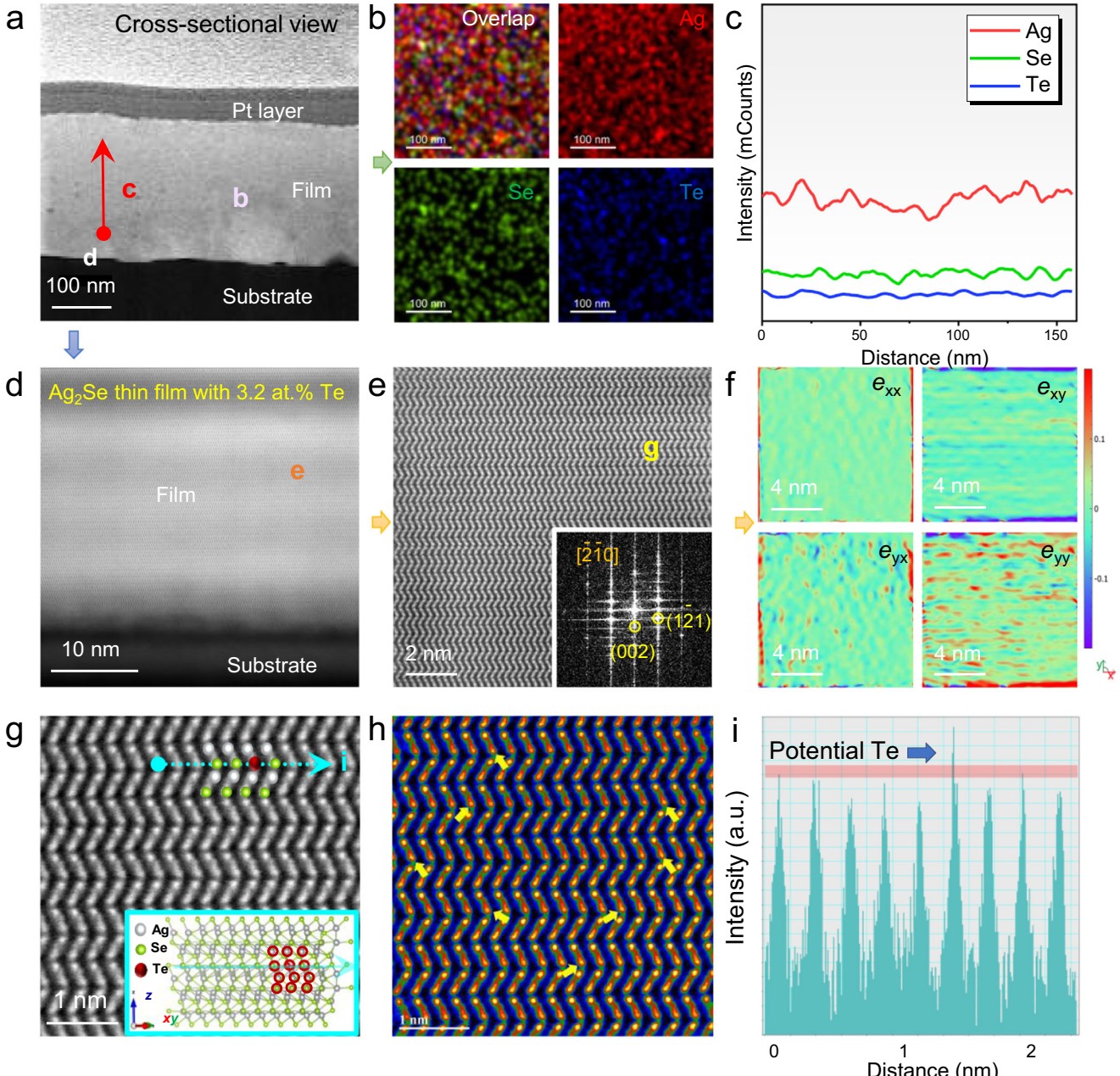

**Fig. 3 | Micro/nanostructural characterizations of the Ag₂Se thin film with 3.2 at.% Te. a** TEM high-angle annular dark field (HAADF) image of the thin-film sample from a cross-sectional view. The sample was prepared by a focused ion beam (FIB) technique. **b** EDS maps for overlap and individual Ag, Se, and Te elements taken from (**a**). **c** EDS line scan taken from (**a**). **d** Spherical aberration-corrected scanning TEM (Cs-STEM) HAADF image taken from (**a**). **e** Magnified Cs-STEM HAADF image taken from (**d**). The inset shows the corresponding fast Fourier transform (FFT) pattern with indexed information. **f** Corresponding strain maps along different directions. **g** High-resolution Cs-STEM HAADF image taken from (**e**). The inset shows the crystal structure of Te-doped Ag₂Se for reference. **h** Filtered Cs-STEM HAADF image to show the contrast difference. The arrow indicates a potential point defect of Te$_{Se}$. **i** Line profile taken from (**g**) to show the potential point defect of Te$_{Se}$.

gradually optimized by increasing the Te doping concentration to achieve peak *ZT* in the thin film.

To interpret the effect of phonon scattering on $\kappa_l$ reduction, the Callaway model is adopted. A decrease $\kappa_l$ is mainly responsible for the reduction in $\kappa$ depending on different phonon scattering phenomena. Debye Callaway model is investigated to clarify the source of reduction in $\kappa_l$[41]:

$$\kappa_l = \frac{4\pi k_B^4 T^3}{v h^3} \int_0^{\frac{\theta_D}{T}} \tau_T \frac{\chi^4 \exp(\chi)}{[\exp(\chi) - 1]^2} d\chi \qquad (1)$$

here $v, \theta_D, \gamma$, and $\tau_T$ stand for sound velocity, Debye temperature, reduced phonon frequency ($\chi = \hbar\omega/k_B T$), and total phonon relaxation time. Remarkably, strong scattering of phonon is originated by alloy element/intrinsic defects and grains that are primarily responsible for the reduction in $\kappa_l$ for Te-doped samples. Herein, the obtained $\kappa_l$ data curves (solid lines) exhibit inconsistency with the experimental results for Te doping concentrations lower than 3.2 at.% (Fig. 5c) at high temperatures, aligning with the trend of $\kappa_l + \kappa_b$ (where $\kappa_l$ is bipolar thermal conductivity) as a function of $T^{-1}$ (Supplementary Fig. 15a). This inconsistency is likely due to the influence of bipolar diffusion on thermal conductivity at high temperatures (Supplementary Fig. 15b).

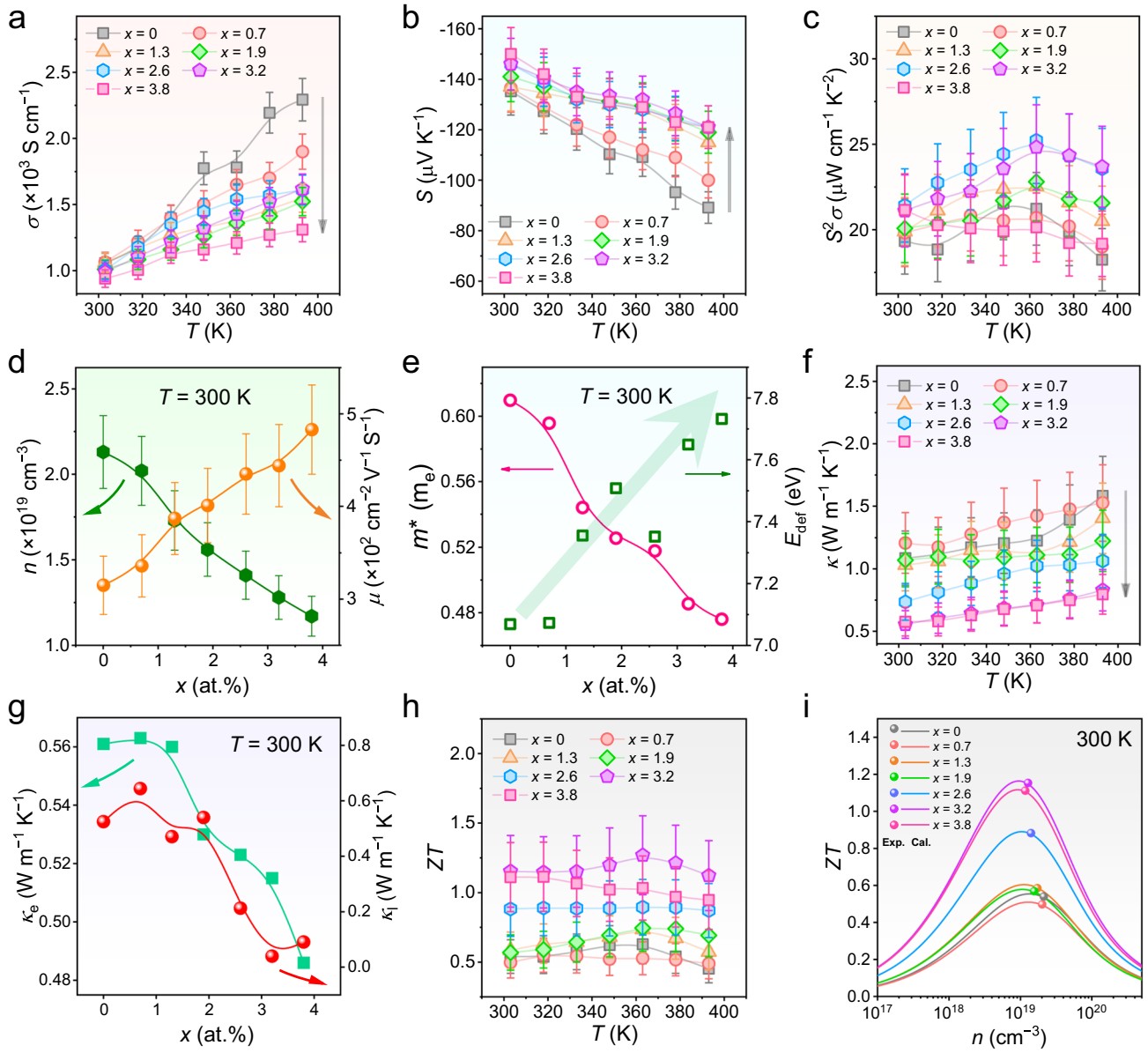

**Fig. 4 | Thermoelectric properties of Ag₂Se thin films with different Te concentrations $x$ ($x = 0, 0.7, 1.3, 1.9, 2.6, 3.2,$ and 3.8 at.%).** Temperature-dependent (**a**) electrical conductivity $\sigma$, (**b**) Seebeck coefficient $S$, and (**c**) determined $S^2\sigma$. **d** Measured $x$-dependent carrier concentration $n$ and mobility $\mu$ at room temperature. **e** Calculated $x$-dependent effective mass $m^*$ and deformation potential $E_{def}$ at room temperature by a single parabolic band (SPB) model. Temperature-dependent (**f**) thermal conductivity $\kappa$. **g** $x$-dependent electronic and lattice thermal conductivities ($\kappa_e$ and $\kappa_l$) at room temperature. **h** Temperature-dependent $ZT$. **i** Comparison of calculated and experimental $ZT$s as a function of $n$. The calculated data are from the SPB model.

As illustrated in Fig. 5d, phonon scattering from grain boundaries, point defects, and/or crystal defects may significantly reduce phonon transport. Simultaneously, the calculated curves/values determine the primary contribution of phonon scattering from impurity/point defects and grain boundaries (Fig. 5e, f). Details regarding the calculations and parameters used in the Debye-Callaway model for all investigated doped specimens are provided in the Supplementary Information.

After Te doping, although the structure of the Ag₂Se film has been optimized in uniformity (overall orientation tends to favor (00$l$)), this does not necessarily mean that its flexibility has significantly improved. In fact, after doping with Te, the flexibility is only slightly improved (Supplementary Fig. 12). Additionally, the increase in $E_{def}$ caused by Te doping may lead to a decrease in lattice variability, potentially resulting in increased film hardness. Therefore, to further enhance the practical value of the as-

obtained high-performance thin film, significant improvements in its flexibility are required. In this work, we designed a protective layer made of organic polymer composites (90% poly (vinyl laurate) and 10% N-methylpyrrolidone), which is applied to the film surface using a spin-coating process. Figure 6a, b compare the measured $\Delta R/R_0$ of the Ag₂Se thin films with 3.2 at.% Te without and with vinyl-laurate-based coatings at different bending radii and bending cycles. The inset in Fig. 6b shows a photo of the flexible thin film. For a better comparison, the vertical axis ranges of both graphs have been adjusted to be consistent. Obviously, after coating the organic polymer composites, the overall flexibility of the thin film was significantly improved. After 1000 bending cycles with a bending radius of 6.3 mm, the performance loss is only 2.5%. Such outstanding flexibility is highly competitive compared with reported works based on inorganic thermoelectric thin films (Table 2)[42–48]. Supplementary Fig. 13 also provides

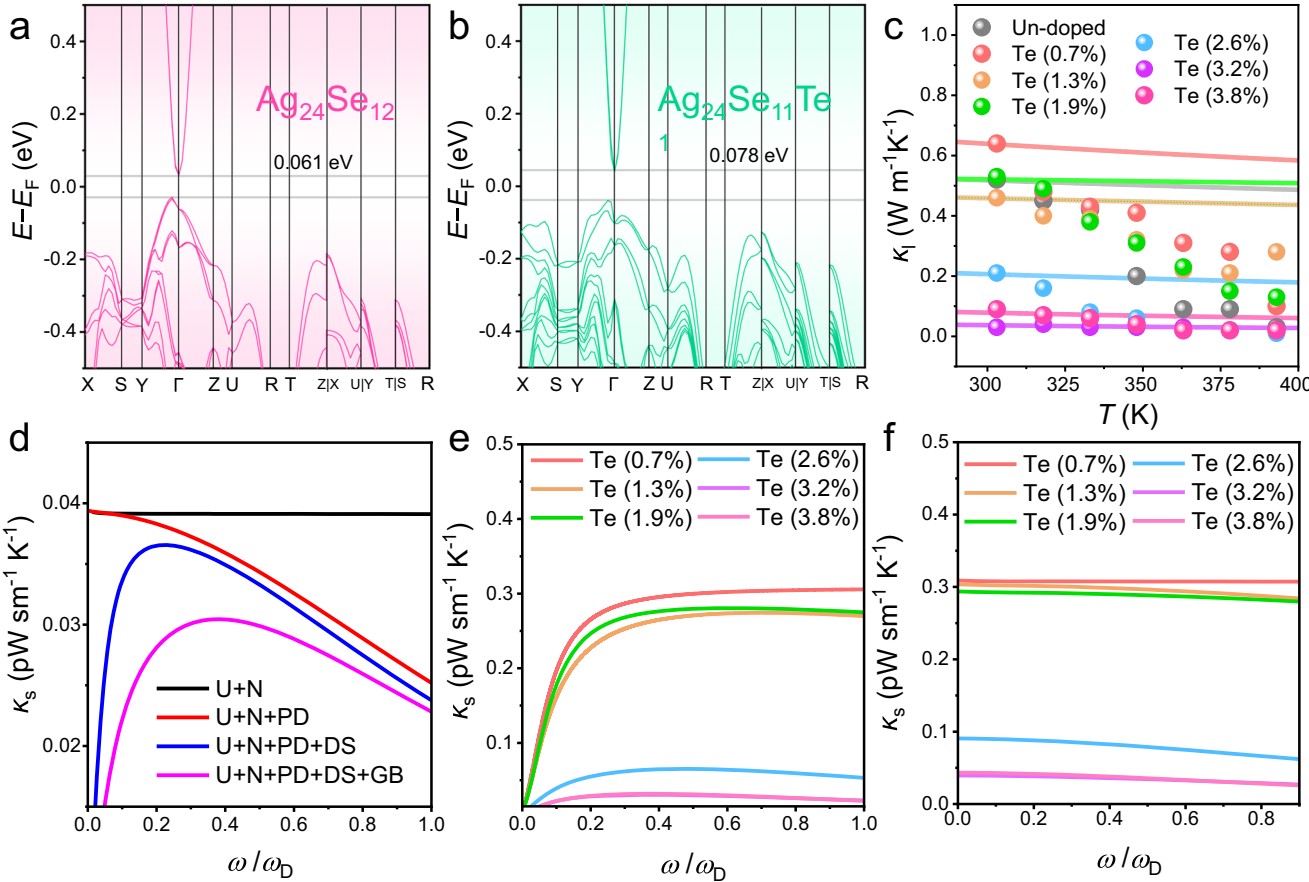

**Fig. 5 | Calculation results of Te-doped Ag₂Se thin films.** Calculated band structures of (**a**) Ag₂Se (Ag₂₄Se₁₂) and (**b**) Te-doped Ag₂Se (Ag₂₄Se₁₁Te₁). (**c**) Measured $\kappa_l$ compared with calculated $\kappa_l$ as a function of temperature. **d** Theoretically derived spectral $\kappa$ ($\kappa_s$) for Ag₂Se thin films with different Te concentrations $x$ ($x = 0$, 0.7, 1.3, 1.9, 2.6, 3.2, and 3.8 at.%), with various phonon scatterings. Here scattering is from phonon-phonon Umklapp- and normal-process (U + N), vacancies/alloy elements (point defects, PD), grain boundaries (GB), and dislocations (DS). **e** Effect of point-defect scattering and (**f**) dependence of grain size on reducing $\kappa_s$ as a function of Te content.

measured $\Delta R/R_0$ of the coated Ag₂Se thin films with 3.2 at.% Te at different bending radii and heating at 95 °C by different periods for 1000 times bending, confirming the high flexibility at higher temperatures.

Based on this novel coated thin film, we designed a facile F-TED to confirm its practicality. To further enhance the output performance of the F-TED, suppressing the contact resistance $\rho_c$ between the thin film and the electrodes is crucial. Therefore, we need to choose appropriate electrode materials to match the thin-film material, which has been rarely reported in the designs of flexible F-TEDs in the past[2,49]. Figure 6c shows measured $\rho_c$ between Ag₂Se thin film with 3.2 at.% Te and different electrode materials including Au, Al, Mo, and Cu. The measurement principle is provided in Supplementary Fig. 14 for reference. We selected Cu as the potential electrode material that possesses the lowest $\rho_c$. Based on this, we assembled the F-TED and evaluated its output performance. The main room-temperature thermo-electric properties of both Sb₂Te₃ and Ag₂Se are provided in Table 3. Figure 6d compares measured open-circuit voltages and output powers of the device as a function of loading current at different $\Delta T$s, and Fig. 6e compares determined output power densities at different $\Delta T$s. The inset shows the photo of a flexible device composed of one pair of Ag₂Se thin film with 3.2 at.% Te as an n-type leg and Sb₂Te₃ thin film as a p-type leg. Remarkably, a high open-circuit voltage of 6 mV, a substantial output power of 65 nW, and a competitive power density of 1.5 mW cm⁻² can be simultaneously achieved at a $\Delta T$ of 20 K[2,49]. The comparison

results are provided in Supplementary Table 3. Figure 6f compares the measured voltage of wearing the F-TED with five p-n legs for sitting, walking, and running as a function of time. The maximum voltage was obtained when wearing the device while running. This is because during running, there is better heat dissipation at the side of the device that is exposed to the air, which helps maintain a larger temperature gradient and thus allows the device to perform better.

In summary, this work reports a record-high $ZT$ of 1.27 at 363 K in polycrystalline Ag₂Se-based thin film with 3.2 at.% Te. Theoretical calculations indicate that Te-doping can reduce the formation energy of the (00$l$) surface, resulting in a strong (00$l$) orientation across the film to enhance its uniformity and in turn carrier mobility. Moreover, Te-doping slightly broadens the bandgap and results in an enhanced Seebeck coefficient and a high power factor of 24.8 μW cm⁻¹ K⁻² by optimizing the carrier concentration. Te-doping at Se sites also introduces Te_Se point defects that effectively scatter phonons, reducing thermal conductivity to 0.71 W m⁻¹ K⁻¹. After 1000 bending cycles with a bending radius of 6.3 mm, the film shows only 2.5% performance loss by coating a protective layer made of organic-polymer-based composites on the film surface, indicating significantly enhanced overall flexibility. Finally, the assembled flexible thermoelectric device using a pair of p-type Sb₂Te₃ and n-type Ag₂Se films achieves an open-circuit voltage of 6 mV, an output power of 65 nW, and a power density of 1.5 mW cm⁻² under a temperature difference of 20 K. This work indicates Te doping can significantly

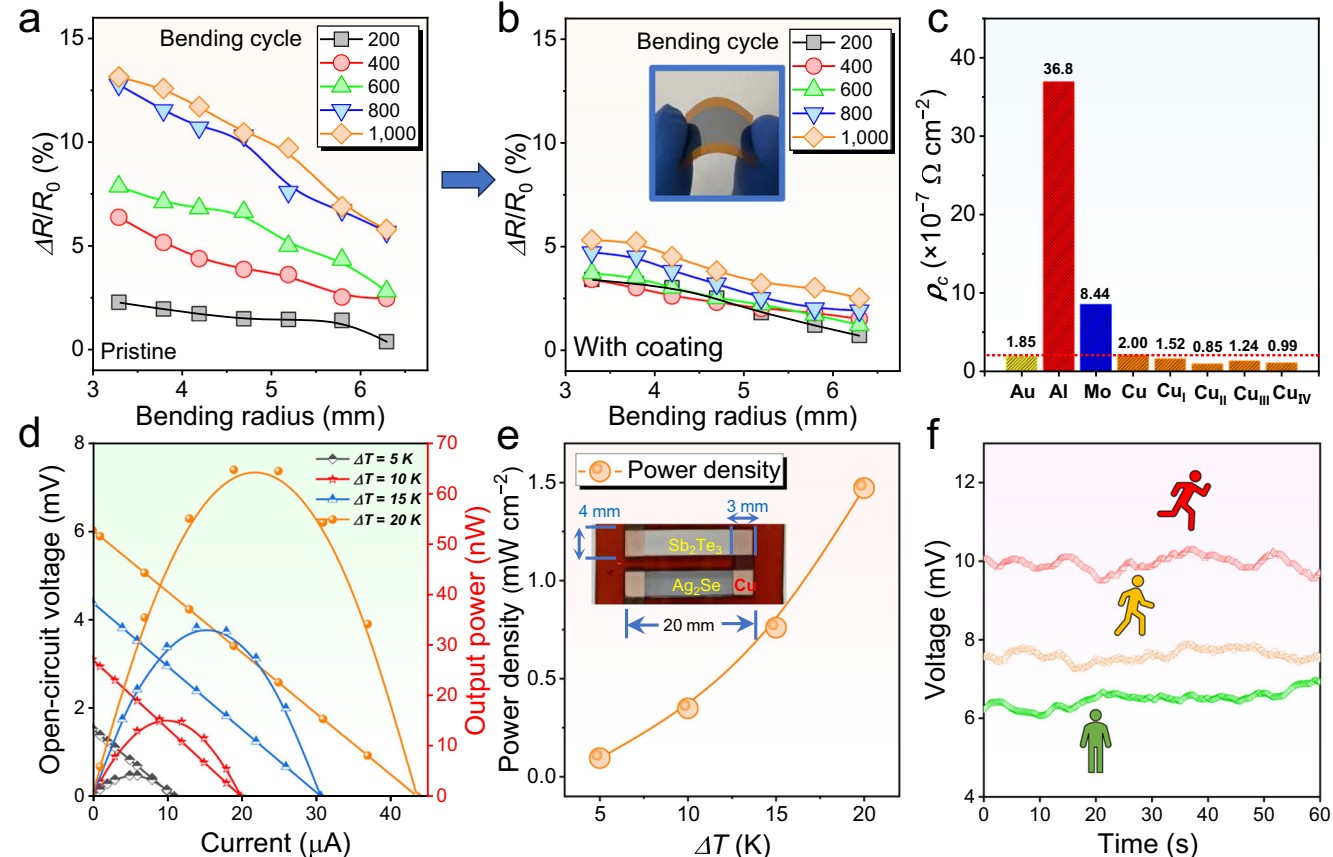

**Fig. 6 | Flexibility of Te-doped Ag$_2$Se thin films and their device performance.** Measured $\Delta R/R_0$ of the Ag$_2$Se thin films with 3.2 at.% Te (**a**) without and (**b**) with a vinyl-laurate-based coating at different bending radii and bending cycles. The inset in (**b**) shows a photo of the flexible thin film. **c** Measured contact resistivity $\rho_c$ between coated Ag$_2$Se thin film with 3.2 at.% Te and different electrode materials. **d** Measured open-circuit voltages and output powers of the device as a function of loading current at different temperature differences ($\Delta Ts$). **e** Determined output power densities at different $\Delta Ts$. The inset shows the photo of a flexible device composed of one pair of coated Ag$_2$Se thin film with 3.2 at.% Te as an n-type leg and coated Sb$_2$Te$_3$ thin film as a p-type leg. **f** Voltage of wearing the as-fabricated device for sitting, walking, and running as a function of time.

improve the thermoelectric potential of Ag$_2$Se films in the application of wearable power generation.

## Methods

### Thin film and device fabrication

The flexible Ag$_2$Se thin films were fabricated on PI substrates by the vacuum thermal co-evaporation method at various temperatures.

**Table 2 | Comparison of flexibility between Ag$_2$Se thin film with 3.2 at.% Te and reported works**

| Materials | Bending cycle | $r$ (mm) | $\Delta R/R_0$ | Ref. |
|---|---|---|---|---|
| Ag$_{0.005}$Bi$_{0.5}$Sb$_{1.5}$Te$_3$ | 1000 | 5 | ~20% | 42 |
| Bi$_2$Te$_3$ | 2000 | 7 | ~30% | 43 |
| Bi$_2$Te$_3$ | 10 | 4 and 60 | ~21% | 44 |
| Bi$_2$Te$_3$/SWCNT | 600 | 4 | ~4% | 45 |
| Bi$_{0.5}$Sb$_{1.5}$Te$_3$ | 1000 | 5 | ~20% | 46 |
| Bi$_2$Te$_3$/CF | 100 | 10 | ~6% | 47 |
| Bi$_{0.4}$Sb$_{1.6}$Te$_3$ + 8 wt% Te | 1000 | 10 | ~2.5% | 48 |
| Ag-doped Bi$_2$Te$_3$ | 2000 | <6 | <4% | 6 |
| Te-doped Ag$_2$Se | 1000 | ~6.3 | ~2.5% | This Work |

The data comprise bending cycles, bending radius $r$, and the increase in normalized resistivity $\Delta R/R_0$ for various inorganic flexible films. In this context, CF stands for cellulose fiber.

After that, a single Ag$_2$Se-Sb$_2$Te$_3$ thermoelectric device was fabricated with a Cu electrode on a PI substrate. Then, an organic coating material, polyvinyl laurate, was employed to enhance the flexibility of the as-fabricated Ag$_2$Se thin films. The experimental details can be found in the Supplementary Information.

### Characterizations and thermoelectric performance evaluation

The details can be found in the Supplementary Information.

### Theoretical calculation

The First-principles calculations and single parabolic (SPB) model was performed to understand the carrier transport performance. The Debye-Callaway model was investigated to clarify the source of reduction of thermal conductivity. More details can be found in the Supplementary Information.

**Table 3 | Thermoelectric properties of Ag$_2$Se thin film with 3.2 at.% Te as an n-type leg and Sb$_2$Te$_3$ thin film as an p-type leg in the flexible thermoelectric device**

| Materials | Type | Thickness (nm) | $S$ ($\mu$V K$^{-1}$) | $\sigma$ (S cm$^{-1}$) | $S^2\sigma$ ($\mu$W cm$^{-1}$ K$^{-2}$) |
|---|---|---|---|---|---|
| Ag$_2$Se thin film with 3.2 at.% Te | $n$ | ~350 | −146.0 | 990.0 | 21.1 |
| Sb$_2$Te$_3$ thin film | $p$ | ~650 | 143.2 | 513.6 | 10.5 |

## Data availability

The data generated in this study are provided in the Supplementary Information/Source Data file. Source data are provided with this paper.

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

## Acknowledgements

This work was supported by the National Natural Science Foundation of China (No. 62274112), Guangdong Basic and Applied Basic Research Foundation (2022A1515010929), and Science and Technology plan project of Shenzhen (JCYJ20220531103601003). ZGC thanks the financial support from the Australian Research Council, and QUT Capacity Building Professor Program, and acknowledges the National Computational Infrastructure, supported by the Australian government, for providing computational resources and service. The authors are thankful for the assistance on STEM-HAADF observation received from the Electron Microscope Center of Shenzhen University. This work was enabled by the use of the Central Analytical Research Facility hosted by the Institute for Future Environments at QUT.

## Author contributions

D.Y. and X.-L.S. contributed equally to this work. Z.-G.C. and Z.H.Z. supervised the project and conceived the idea. D.Y. and X.-L.S. designed the experiments and wrote the manuscript. D.Y. carried out the synthesis of materials. M.L., X.-L.S., W.L., H.M. and X.Z. undertook the theoretical work. M.N., A.M., S.C., Y.C., L.F., G.-X.L., P.F. undertook the thermoelectric performance evaluation. All the authors discussed the results and commented on the manuscript. All authors have given approval to the final version of the manuscript.

## Competing interests

The authors declare no competing interests.
