## [Peer Review File · Nature Communications]

REVIEWER COMMENTS

Reviewer #1 (Remarks to the Author):

The authors reported on the Te-doped Ag₂Se films and their excellent thermoelectric performance and flexibility. By the Te doping, crystal orientation of the Ag₂Se was successfully controlled. The Te doping effect on thermoelectric performance of the Ag₂Se films were systematically studied in depth. Flexibility test of the Ag₂Se was also successfully conducted. This study is fascinating and very interesting. I have one major comment and one minor comment:

In this study, in-plane Seebeck coefficient and in-plane electrical conductivity were measured. However, measurement direction of thermal conductivity is unclear. How did the authors measure and extract the in-plane thermal conductivity of the thermoelectric film constrained on a substrate. Previously, X. Crispin group and L. Shi group reported that in-plane thermal conductivity measured by 3 omega method can be incorrect. Use of suspended microdevices is recommended for the measurement of in-plane thermal conductivity of the films constrained on a substrate. [Advanced Materials 27 (2015) 2101] Use of the suspended microdevices is strongly recommended for the measurement of the in-plane thermal conductivity and zT.

(Minor comment) In the title, "wearable" should be replaced to "flexible".

Reviewer #2 (Remarks to the Author):

The manuscript (NCOMMS-23-53082-T) by Yang et al. presents the fabrication of flexible Ag₂Se thin films for wearable power generators. They deposited their films on PI flexible substrate by a vacuum thermal co-evaporation method and improved the crystallographic texture by doping Te. The power factor and ZT were impressively high, and the films are very stable after subjecting to cyclic bending tests. Overall, I think the experiments were rationally well designed and the results are scientifically sound. The manuscript can be accepted for publication after minor revision according to my comments below.

1. Figure 4(a) – (i) should be rearranged in a correct alphabetical order.
2. I feel like there are too many details for the Callaway model analysis from line 303 to 355. Some of the details and equations can be moved to supplementary, which will make the manuscript neat and clean. The flow and readability will be improved.
3. Could you add some comments why the experiments and models in Figure 5c fit well for only Te 3.2 and 3.8%? For other samples, they do not seem to match very much.
4. There are 50 exact references for both main paper and the supporting information, but they are different groups of references, which caused me confusion in the first place. Maybe, the authors can add some comments in the supporting information to clarify that the references are the different set.

Reviewer #3 (Remarks to the Author):

In this manuscript, Yang et al reported the orientated Ag₂Se film with Te element doping, exhibiting high thermoelectric (TE) performance near room temperature. With the flexible organic substrate, flexible and wearable devices of Ag₂Se was fabricated and achieved an output power density of 1.5 mWcm⁻². I think this work is helpful to the flexible thermoelectrics. There are some concerns regarding some specific issues.

[1] The average thickness of Ag₂Se films is marked as 350 nm in the Supporting information. However, the film thickness mentioned in device fabrication in Table 3 is 450 nm. Also, for the transport property measurement, do all the samples show a thickness of ~350 nm? Possible variation in thickness should be indicated.

[2] In Figure 3h, it is argued that the contrast difference can indicate the potential point defect of TeSe. But the contrast is not significant, and other places also show the same contrast marked in Figure 3h. Please further check the relevant discussion.

[3] The EDS map in Figure S9 shows the inhomogeneous distribution of Ag, Se, and Te elements. Except for TeSe defects, are there any other defects?

[4] In Figure 4e, the effective mass of a series of Ag₂Se samples is larger than 0.5 me. This value is much larger than previously reported Ag₂Se samples at around 0.2 me (Journal of Materiomics, 2023, 4, 754). Please explain this large difference.

[5] The calculated lattice thermal conductivity decreases dramatically with increasing Te content, even reaching zero. With only 4 at.% Te, the lattice thermal conductivity has a reduction of 0.6 Wm⁻¹K⁻¹. Here both the measurement of k for thin films and the calculation of k_L for highly conductive materials are challenging tasks. I suggest double-checking both the measurement and calculation methods. Also, perhaps the theoretical minimum lattice thermal conductivity can be calculated as a reference.

[6] According to Table 3, the total electrical resistance of the single leg is supposed to be small. However, according to the slope in Figure 6d, the device's resistance is larger than 100 Ω. Please check the testing methods or fabrication process, and explain the resistance difference. The output performance of the device should be compared with other state-of-the-art flexible devices to show the possible superiority of this work.

[7] Why select Sb₂Te₃ as the other leg?

[8] The calculated surface energy of (001) is reported as 0.002~0.004 eV/A², which seems quite small? How about other materials?

Response to Reviewers

Reviewer #1:

The authors reported on the Te-doped Ag₂Se films and their excellent thermoelectric performance and flexibility. By the Te doping, crystal orientation of the Ag₂Se was successfully controlled. The Te doping effect on thermoelectric performance of the Ag₂Se films were systematically studied in depth. Flexibility test of the Ag₂Se was also successfully conducted. This study is fascinating and very interesting. I have one major comment and one minor comment:

Author reply: We appreciate the positive comments and constructive suggestions.

Comment 1: In this study, in-plane Seebeck coefficient and in-plane electrical conductivity were measured. However, measurement direction of thermal conductivity is unclear. How did the authors measure and extract the in-plane thermal conductivity of the thermoelectric film constrained on a substrate. Previously, X. Crispin group and L. Shi group reported that in-plane thermal conductivity measured by 3 omega method can be incorrect. Use of suspended microdevices is recommended for the measurement of in-plane thermal conductivity of the films constrained on a substrate. [Advanced Materials 27 (2015) 2101] Use of the suspended microdevices is strongly recommended for the measurement of the in-plane thermal conductivity and zT .

Author reply: In this study, we determined the Seebeck coefficient and electrical conductivity using the SBA 458-NETZSCH Seebeck Coefficient and Electrical Conductivity Analyzer. Additionally, thermal conductivity was measured by the 3 ω method using a thin film comprehensive physical property analyzer (TFA-LINSEES). All measurements were conducted along the in-plane direction. According to the performance report provided by LINSEES, a manufacturer based in Germany, the in-

plane thermal conductivity measurement is subject to an error range of $\pm 7\%$ to $\pm 10\%$.

The measurement process and method details are outlined as follows: We collaborated with a professional testing company with instruments specialized in analyzing the thermal properties of films: TFA-LINSEIS (www.linseis.com) and utilized a 2nd generation measurement chip, depicted in **Figure R1**. This chip integrates two suspended membrane setups ¹, employing the Völklein geometry ^{2,3}, for in-plane thermal conductivity measurements. The chosen method has been widely adopted in numerous articles for measuring in-plane thermal conductivity ⁴⁻⁸.

Figure R1. (a) Illustration of the chip cross-section after cutting. (b) Schematic cross-sectional view of the larger membrane, featuring an embedded thermometer. (c) Top view of the dual-membrane configuration, showcasing heating stripes and a resistance thermometer. $I_{M1,A}$, $I_{M1,B}$, $V_{M1,A}$, and $V_{M1,B}$ denote the current and voltage connection pads for the two heaters on membranes 1 and 2. Additionally, I_{RT} and V_{RT} represent the connection pads for the resistance thermometer, and $V_{TH,H}$ is the pad for the hot contact of the thermovoltage measurement ¹.

For the measurements, two heating stripes, each with a width less than $5 \mu\text{m}$, are applied to the front side of two free-standing Si_3N_4 membranes with distinct geometries. To enhance accuracy, a

platinum rim encompassing the membranes serves as a heat sink. To ensure precision in the measurements, the product of the film thermal conductivity (λ_f) and the film thickness (d_f) should be equal to or greater than the corresponding product ($\lambda_s d_s$) of the substrate. ²:

$$\lambda_f d_f \geq \lambda_s d_s \quad (1)$$

In this chip, the passivation layer has a thickness of 30 nm, and its measured thermal conductivity at room temperature is $1.4 \text{ W m}^{-1} \text{ K}^{-1}$. The addition of this passivation layer results in only a 15% increase in the thermal conductance of the measurement setup. The flat surface of the chip enables the deposition of a thin film with uniform thickness, which can be achieved through methods such as spin coating, thermal evaporation, or magnetron sputtering.

To address the potential error from heat loss attributed to radiation, similar to the correction proposed for DC measurements by Völklein et al. ², the measurement chip utilizes a dual membrane correction along with the 3ω method. This approach ensures accurate measurement of in-plane thermal conductivity under quasi-steady-state conditions.

For the measurement, an AC current $I = I_0 \cos(\omega t)$ is applied to the heating stripes, inducing a temperature increase in the membrane represented by $\Delta T = \Delta T_0 \cos(2\omega t + \varphi)$. This leads to an oscillation in the resistance of the stripe $R = R_0 (1 + \alpha \Delta T)$ at the angular frequency of 2ω . The phase shift φ , contingent on both the heater's geometry and the underlying materials, is associated with the frequency. By measuring the voltage drop across the heater using Ohm's law, an amplitude-modulated signal is obtained, featuring a minor component at the third harmonic 3ω . This component can be isolated using a lock-in amplifier.

By solving the one-dimensional thermal heat diffusion equation within the membrane area and considering the specified boundary conditions, it can be demonstrated that the general expression for

the amplitude of the 3ω oscillation ($V_{3\omega}$) is given by ⁹:

$$|V_{3\omega}| = \frac{\alpha R_0^2 I_0^3}{4K_P \sqrt{1 + \omega^2 \left(4\tau^2 + \frac{w^4}{24D^2} + \frac{\tau w^2}{3D} \right)}} \quad (2)$$

with the temperature coefficient of resistance (α), the unloaded resistance of the heater (R_0), the width of the membrane (w), the length of the membrane (l), and the thermal diffusivity of the sample plus membrane setup ($D = \lambda/\rho_m c$, where ρ_m is the mass density and c is the specific heat capacity), the angular frequency is denoted as ω . K_P is defined as $2\lambda t l/w$, and the thermal relaxation time is $\tau = \dot{C}/K_P$. The temperature coefficient of resistance can be determined by linear or cubic fitting of the measured resistance over the temperature of the heating stripes, employing a small measuring current in conjunction with the thermocouple under the chip. The product λt and the total specific heat \dot{C} of the membrane setup can be extracted by fitting the measured 3ω voltage versus frequency using **Eq. (2)**. When utilizing low frequencies, the measurement can be performed under quasi-steady-state conditions, rendering the ω term negligible and resulting in a constant $V_{3\omega}$. In this case, **Eq. (2)** can be expressed in a simpler form:

$$|V_{3\omega}| = \frac{\alpha R_0^2 I_0^3}{4K_P} \quad (3)$$

The sample's thermal conductivity λ_s can be calculated from a differential measurement, using the values of the empty measurement setup (λ_m, d_m) and sample thickness d_s with

$$\lambda_s = \frac{\lambda d - \lambda_m d_m}{d_s} \quad (4)$$

The dual membrane correction employed in this study can be implemented post-measurement by taking the ratio of the thermal conductance of the two membranes,

$$\frac{G_{CR1}}{G_{CR2}} = \frac{l_{M1} \mu \coth(\mu \frac{w_1}{2})}{l_{M2} \mu \coth(\mu \frac{w_2}{2})} \quad (5)$$

with

$$\mu^2 = \frac{8\varepsilon_s \gamma T_0^3}{\lambda t} \quad (6)$$

where γ is the Stefan-Boltzmann constant and ε_s is the emissivity of the sample. When the measurement is conducted at very low frequencies, the phase shift tends toward zero, and the ω term becomes negligible⁹, resulting in a quasi-steady-state measurement. The thermal conductance (G_{CR}) of the membrane, accounting for both heat conduction and heat radiation, is then expressed as:

$$G_{CRi} = 2\lambda t l_{Mi} \mu \coth\left(\mu \frac{w_i}{2}\right) = \frac{N_{0i}}{\Delta T_{Mi}} \quad (7)$$

where $\Delta T_{Mi} = 2|V_{3\omega i}| / \square_i R_{0i} I_{0i}$ is the amplitude of the mean temperature rise and $N_{0i} = R_{0i} I_{0i}^2 / 2$ is the mean dissipated heating power with $i=1, 2$ for the big or small membrane, respectively. The parameter μ can be determined by a numerical calculation of the zero of **Eq. (7)**,

$$G_{CR1} l_{M2} \coth\left(\mu \frac{w_2}{2}\right) - G_{CR2} l_{M1} \coth\left(\mu \frac{w_1}{2}\right) \quad (8)$$

after this, the corrected thermal conductivity and the emissivity of the sample can be calculated using **Eqs. (8), (3), and (9)**,

$$\lambda d = \frac{G_{CR1}}{2l_{M1}\mu \coth\left(\mu \frac{w_1}{2}\right)} = \frac{G_{CR2}}{2l_{M2}\mu \coth\left(\mu \frac{w_2}{2}\right)} \quad (9)$$

$$\varepsilon_s = \frac{\mu^2 \lambda d}{8\gamma T_0^3} \quad (10)$$

In accordance with the aforementioned explanation, this test effectively eliminates the influence of the substrate, enabling precise measurement of the thermal conductivity of thin films. Consequently, we deposited our thin films on the standard chip utilized by this instrument for in-plane thermal conductivity measurements. The results were consistently measured multiple times to ensure the accuracy of the data.

Indeed, it is noteworthy that both Prof. X. Crispin's group and Prof. L. Shi's group have provided an excellent method for accurately measuring the in-plane thermal conductivity of materials¹⁰. These suspended microdevices effectively mitigate errors arising from substrate heat loss and facilitate precise data analysis. However, the method necessitates materials that can exist as independent, free-

standing structures. In our attempts, we endeavored to create a free-standing inorganic Ag₂Se thin film by depositing it on a NaCl substrate and subsequently dissolving the NaCl. Unfortunately, the resulting Ag₂Se thin films were too thin, approximately 350 nm, to achieve free-standing status. Furthermore, our group, along with our collaborating institution, is still in the process of mastering this measurement technique. Consequently, we were unable to employ this method in the current work.

Based on your comments, the relevant sections in the manuscript and supplementary information have been revised as follows:

Revised manuscript (Page 12): “This is partly attributed to the fact that thermal-conductivity measurements of thin films typically have a significant margin of error. Although methods for measuring the in-plane κ of polymer thin films have been reported¹⁰, the current technology for testing the in-plane κ of inorganic thin films is not yet fully mature¹¹”.

Revised supplementary information (Page 3) “The in-plane thermal conductivity κ of the thin film in this work, deposited on a commercial flat chip, was determined by using the 3 ω method through a thin film comprehensive physical property analyzer (TFA-LINSEES).”

Comment 2: In the title, "wearable" should be replaced to "flexible".

Author reply: The title has been revised as “Flexible power generators by Ag₂Se thin films with record-high thermoelectric performance” (**Revised manuscript, Page 1**).

Reviewer #2:

The manuscript (NCOMMS-23-53082-T) by Yang et al. presents the fabrication of flexible Ag₂Se thin films for wearable power generators. They deposited their films on PI flexible substrate by a vacuum thermal co-evaporation method and improved the crystallographic texture by doping Te. The power factor and ZT were impressively high, and the films are very stable after subjecting to cyclic bending tests. Overall, I think the experiments were rationally well designed and the results are scientifically sound. The manuscript can be accepted for publication after minor revision according to my comments below.

Author reply: We appreciate the positive feedback and constructive suggestions, both of which are valuable in further enhancing our manuscript.

Comment 1: Figure 4(a) – (i) should be rearranged in a correct alphabetical order.

Author reply: Figure 4 has been rearranged in the correct alphabetical order.

Comment 2: I feel like there are too many details for the Callaway model analysis from line 303 to 355. Some of the details and equations can be moved to supplementary, which will make the manuscript neat and clean. The flow and readability will be improved.

Author reply: According to the suggestions, we have relocated certain details to the supplementary information (**Page 5**). The modifications in both the manuscript and supplementary information are indicated in red.

Comment 3: Could you add some comments why the experiments and models in Figure 5c fit well

for only Te 3.2 and 3.8%? For other samples, they do not seem to match very much.

Author reply: Generally, thermal conductivity κ is the sum of lattice thermal conductivity (κ_l), electronic thermal conductivity (κ_e), and bipolar thermal conductivity (κ_b), depicted by the relation: $\kappa = \kappa_l + \kappa_e + \kappa_b$. The significant reduction in κ_l is attributed to phonon scattering from grain boundaries, point defects, and/or crystal defects, as illustrated in **Figure 5**.

Concerning the temperature dependence of the data, κ_l for samples ($x < 2.6\%$) exhibits a pronounced reduction with increasing temperature. For instance, κ_l decreases from $0.52 \text{ W m}^{-1} \text{ K}^{-1}$ (at $T = 303 \text{ K}$) to $0.03 \text{ W m}^{-1} \text{ K}^{-1}$ (at $T = 393 \text{ K}$) for the pristine sample ($x = 0.0\%$), from $0.46 \text{ W m}^{-1} \text{ K}^{-1}$ (at $T = 303 \text{ K}$) to $0.28 \text{ W m}^{-1} \text{ K}^{-1}$ (at $T = 393 \text{ K}$) for $x = 1.3\%$, and from $0.53 \text{ W m}^{-1} \text{ K}^{-1}$ (at $T = 303 \text{ K}$) to $0.13 \text{ W m}^{-1} \text{ K}^{-1}$ (at $T = 393 \text{ K}$) for $x = 1.9\%$. However, at higher doping levels ($x > 2.6\%$), κ_l values decrease more slightly. For example, the κ_l values decrease from $0.21 \text{ W m}^{-1} \text{ K}^{-1}$ (at $T = 303 \text{ K}$) to $0.01 \text{ W m}^{-1} \text{ K}^{-1}$ (at $T = 393 \text{ K}$) for $x = 2.6\%$, from $0.03 \text{ W m}^{-1} \text{ K}^{-1}$ (at $T = 303 \text{ K}$) to $0.02 \text{ W m}^{-1} \text{ K}^{-1}$ (at $T = 393 \text{ K}$) for $x = 3.2 \%$, and from $0.09 \text{ W m}^{-1} \text{ K}^{-1}$ (at $T = 303 \text{ K}$) to $0.02 \text{ W m}^{-1} \text{ K}^{-1}$ (at $T = 393 \text{ K}$) for $x = 3.8 \%$.

The deviation between theoretically derived κ_l and experimental κ_l can be attributed to the bipolar effect. In narrow-gap semiconductors at room temperature, both holes and electrons contribute to transport. To clarify the doping-dependent contribution of κ_b at high temperatures, the κ_b is separated from the κ using the method proposed by Kitagawa *et al.*¹². The difference, $\kappa_l + \kappa_b$, as a function of T^{-1} for the doped samples is shown in **Figure R2(a)**. Acoustic phonon scattering is predominant at low temperatures at low doping content, making the $\kappa_l + \kappa_b$ equal to the κ_l , which is proportional to T^{-1} . As the temperature increases, the $\kappa_l + \kappa_b$ gradually deviates from this linear relationship, indicating the onset of bipolar diffusion contribution to the κ at high temperatures. The κ_l is estimated by

extrapolating the linear relationship between κ_1 and T^{-1} , as indicated by the dotted line in **Figure R2(a)**.

With increasing the doping content ($x > 2.6\%$), the κ_b shows a decreasing trend, mainly due to the trapping of minority carriers within the intensive nanoscale regions inside the doped samples. The reduction in κ_b results in the κ drop by reducing the κ_1 and suppressing the κ_b . Another reason for the model's misfit to the experimental data at high temperatures for doped samples lies in the different scattering parameters exhibiting their effect in the temperature range in Ag_2Se .

Furthermore, the defined dominating scattering mechanisms for κ_1 variation values, the proposed pre-factors Z_2 and Z_3 (corresponding to the relaxation time related to scattering from vacancies/alloy elements (point defects, PD), and grain boundaries (GB)), show a significant reduction compared to τ for the undoped Ag_2Se system due to the intense scattering effects under alloying or nanoscale grain size. The Z_2 parameter increases with the increasing level of Te doping in Ag_2Se , indicating an increase in the anharmonicity of Se–Te and a decrease in Ag–Se chemical bonding. Weak chemical bonding of Ag–Se indicates the persistence of lattice anharmonicity. The weak chemical bonding between Ag and Se atoms enables Ag ions to move freely around the equilibrium position, leading to a high atomic displacement parameter of Ag atoms and forming low-frequency optical modes. The deviation from parabolic behavior is a clear indicator of weak chemical bonding of Ag–Se, indicating the persistence of lattice anharmonicity, as given by an anharmonic approach relation of $u = (-3B)/AK_bT$ (temperature-dependent, where u is the deviation, A and B are the potential positions).

Figure R2. (a) The dotted line represents a linear fit to the lattice thermal conductivity (κ_l) at low temperatures. (b) Bipolar thermal conductivity (κ_b) for Ag_2Se thin films with various Te concentrations ($x = 0, 0.7, 1.3, 1.9, 2.6, 3.2,$ and 3.8 at.%).

According to the suggestions, we have revised the relevant descriptions as follows: “Herein, the obtained κ_l data curves (solid lines) exhibit inconsistency with the experimental results for Te doping concentrations lower than 3.2 at.% (Figure 5c) at high temperatures, aligning with the trend of $\kappa_l + \kappa_b$ (where κ_l is bipolar thermal conductivity) as a function of T^{-1} (Supplementary Figure 15a). This inconsistency is likely due to the influence of bipolar diffusion on the κ at high temperatures (Supplementary Figure 15b). As illustrated in Figure 5d, phonon scattering from grain boundaries, point defects, and/or crystal defects may significantly reduce phonon transport. Simultaneously, the calculated curves/values determine the primary contribution of phonon scattering from impurity/point defects and grain boundaries (Figures 5e-f). Details regarding the calculations and parameters used in the Debye-Callaway model for all investigated doped specimens are provided in the Supplementary Information” (Manuscript Page 13).

Comment 4: There are 50 exact references for both main paper and the supporting information, but

they are different groups of references, which caused me confusion in the first place. Maybe, the authors can add some comments in the supporting information to clarify that the references are the different set.

Author reply: The relevant information has been revised in the Supplementary Information under the section “References (Supplementary Information)”.

Reviewer #3:

In this manuscript, Yang et al reported the orientated Ag₂Se film with Te element doping, exhibiting high thermoelectric (TE) performance near room temperature. With the flexible organic substrate, flexible and wearable devices of Ag₂Se was fabricated and achieved an output power density of 1.5 mW cm⁻². I think this work is helpful to the flexible thermoelectrics. There are some concerns regarding some specific issues.

Author reply: We appreciate the positive feedback and constructive suggestions, both of which are very helpful in further improving our manuscript.

Comment 1: The average thickness of Ag₂Se films is marked as 350 nm in the Supporting information. However, the film thickness mentioned in device fabrication in Table 3 is 450 nm. Also, for the transport property measurement, do all the samples show a thickness of ~350 nm? Possible variation in thickness should be indicated.

Author reply: The deposition processes for all thin films and devices are identical. Throughout the thin film deposition process, we controlled the thickness using a crystal oscillator and confirmed it post-deposition with a mechanical probe profilometer (Dektak XT, Bruker). We have updated the thickness of Ag₂Se in **Table 3** to be approximately 350 nm.

Comment 2: In Figure 3h, it is argued that the contrast difference can indicate the potential point defect of TeSe. But the contrast is not significant, and other places also show the same contrast marked in Figure 3h. Please further check the relevant discussion.

Author reply: Substitutional doping of Te atoms on Se sites is the most plausible defect in our Te-

doped Ag_2Se system due to the similar atom sizes (Ag 172 pm, Se 120 pm, and Te 140 pm) and electronegativities (Ag 1.93, Se 2.55, and Te 2.1). In this study, we detected Te_{Se} defects by using TEM and found that the energy band calculation based on Te substitution was consistent with the experimental results. This indicates the presence of Te_{Se} defects. To better highlight potential defect locations, we have modified the image's color based on the comments, as shown in **Figure R3**. Additionally, **Figure 3h** in the manuscript has been replaced with **Figure R3**.

Figure R3. Filtered Cs-STEM HAADF image to show the contrast difference. The arrows indicate potential point defects of Te_{Se} .

Comment 3: The EDS map in Figure S9 shows the inhomogeneous distribution of Ag, Se, and Te elements. Except for Te_{Se} defects, are there any other defects?

Author reply: Defects come in various types and play a crucial role in semiconductor materials. In the context of our Te-doped Ag_2Se system, defects can primarily occur through substitutional or interstitial sites due to the presence of impurity Te atoms¹³. In our previous analysis, Te_{Se} defects were experimentally observed and identified, but there was no significant evidence of interstitial defects.

According to the suggestions, we conducted a reanalysis of our data by enlarging XRD patterns. Typically, XRD refinement is performed to analyze crystal constants. However, our doped thin films exhibit a highly oriented growth feature, which can impact XRD refinement results and lead to inaccurate analysis¹⁴, as shown in **Figure R4**. The XRD peaks have shifted very slightly toward higher angles after Te doping, indicating that the possibility of interstitial defects is very small.

Figure R4. Enlarged XRD peaks of (002) and (004) crystal planes.

Comment 4: In Figure 4e, the effective mass of a series of Ag₂Se samples is larger than 0.5 me. This value is much larger than previously reported Ag₂Se samples at around 0.2 me (Journal of Materiomics, 2023, 4, 754). Please explain this large difference.

Author reply: The effective mass m^* is a crucial physical quantity that describes the electronic structure of materials. To determine the m^* by the single parabolic band (SPB) model, the reduced

electrochemical potential η should be calculated from the magnitude of the Seebeck coefficient S first
15.

$$S = \frac{K}{e} \left(\frac{2F_1}{F_0} - \eta \right) \quad (11)$$

where K is the Boltzmann constant, F_i is the Fermi integral, $F_i(\eta) = \int_0^\infty \frac{x^i dx}{1 + \exp(x - \eta)}$. Then estimate m^* using η , the temperature T , and the carrier concentration n :

$$n = 4\pi \left(\frac{2m^*KT}{h^2} \right)^{3/2} F_{1/2} \quad (12)$$

The Hall carrier concentration (n_H) is related to the chemical carrier concentration n via $n_H = n/r_H$, where the Hall factor r_H for acoustic phonon scattering is given by:

$$r_H = \frac{3}{2} F_{1/2} \frac{F_{-1/2}}{2F_0^2} \quad (13)$$

According to the analysis above, the m^* is directly proportional to the n_H . Lin et al. reported that Ag_2Se single crystals have an atomic ratio of Ag/Se of ~ 1.94 with an n_H of $\sim 10^{18} \text{ cm}^{-3}$. In our case, the Ag_2Se thin films we obtained have an atomic ratio of Ag/Se of ~ 2.02 with an n_H of $\sim 10^{19} \text{ cm}^{-3}$. This results in a difference of about 4 times in the calculation of the m^* .

The related information has been revised as: “As observed, the n values are higher than those reported for Ag_2Se single crystals ¹⁶, and they can be progressively optimized by increasing the Te doping concentration to achieve peak $S^2\sigma$ in the thin film”. (**Revised manuscript, Page11**)

Comment 5: The calculated lattice thermal conductivity decreases dramatically with increasing Te content, even reaching zero. With only 4 at.% Te, the lattice thermal conductivity has a reduction of $0.6 \text{ W m}^{-1} \text{ K}^{-1}$. Here both the measurement of k for thin films and the calculation of κ_l for highly conductive materials are challenging tasks. I suggest double-checking both the measurement and calculation methods. Also, perhaps the theoretical minimum lattice thermal conductivity can be

calculated as a reference.

Author reply: According to the suggestions, we have thoroughly reviewed the measurement data and calculation methods, and we can confidently confirm that there are no mistakes. As reported by Wang et al.¹⁷, the theoretical minimum κ_1 of Ag-Se-based monolayers can be very close to zero when the mean free path is smaller than 100 nm¹⁷, indicating that this material system can exhibit ultralow κ under appropriate conditions. In fact, different κ values with lower than 0.3 W m⁻¹ K⁻¹ (even as low as 0.1 W m⁻¹ K⁻¹) have been reported in many pure Ag₂Se bulk materials¹⁸⁻²², confirming the theoretical inference that a judicious regulation can yield Ag₂Se with extremely low κ_1 . Meanwhile, similar results have also been reported for Ag₂Se thin films by Marisol Martin-Gonzalez et al.²³ and Cai et al.²⁴. Therefore, all theoretical and experimental results have confirmed that it is possible to obtain extremely low κ_1 in Ag₂Se²⁵. Comparatively, the pure Ag₂Se thin film prepared in this work has a measured κ and κ_1 of ~1.1 W m⁻¹ K⁻¹ and 0.5 W m⁻¹ K⁻¹. This falls within the range of theoretical values (0.9 W m⁻¹ K⁻¹ to 1.3 W m⁻¹ K⁻¹) and is slightly higher than many reported experimental values, including those for bulk and thin films.¹⁹⁻²⁴.

To interpret the achieved low κ_1 after Te doping, the Callaway model was employed in this work. The reduced κ_1 arises from the solid solution and nanoscale size effects. The solid solution introduces an increased concentration of Te_{Se} point defects in the lattice. Additionally, the differences in mass, size, and bonding forces between Se and Te atoms with silver atoms create a strong scattering effect on phonons, leading to a reduction in the κ_1 . The lowest κ was obtained for highly doped samples due to the following factors: (1) their low values of n and m^* (Figures 4d & e). The κ of Ag₂Se is notably more sensitive to changes in the carrier spectrum compared to the phonon spectrum. This underscores the critical importance of tuning the defect structure of Ag₂Se, which governs carrier transport, to

enhance both power factor (PF) and reduce κ ; (2) weak chemical bonding of Ag-Se, indicating the persistence of lattice anharmonicity. The weak chemical bonding between Ag and Se atoms allows Ag ions to move freely around the equilibrium position, resulting in a high atomic displacement parameter of Ag atoms and the formation of low-frequency optical modes; and (3) Ag_2Se exhibiting a low average sound velocity of 1475 m/s compared with other thermoelectric materials. The increased intensity of Te_{Se} point defects is responsible for the scattering of phonons with shorter wavelengths.

Furthermore, to calculate the κ_l , the κ_e must be initially estimated using the formula $\kappa_e = L\sigma T$, where L is the Lorenz number. Obtaining an accurate Lorenz constant is challenging. Consequently, in various reports, a wide range of Lorenz constant values is employed, including literature values, calculated values, or estimated values. This variability can introduce substantial errors in electronic thermal conductivity and, subsequently, result in significant discrepancies in lattice thermal conductivity. Therefore, the objective of presenting the κ_l in this work is not to emphasize its precise value but to illustrate that the κ_l of this material significantly decreases after Te doping. This observation aligns with the theoretical analysis and experimental results, as well as with the majority of other reported findings.

Comment 6: According to Table 3, the total electrical resistance of the single leg is supposed to be small. However, according to the slope in Figure 6d, the device's resistance is larger than 100 Ω . Please check the testing methods or fabrication process, and explain the resistance difference. The output performance of the device should be compared with other state-of-the-art flexible devices to show the possible superiority of this work.

Author reply: Upon reviewing your comments, we have carefully examined the electrical resistance

of the device. According to the formula $R = \rho l/A$, where R is resistance, ρ is resistivity, l is the length along the current, and A is the cross-sectional area perpendicular to the current direction, the resistance of our thin films should be on the order of $10^2 \Omega$. Despite the high electrical conductivity of our films, their thickness is only in the range of hundreds of nanometers. Therefore, when considering contact resistance, the total resistance is not negligible, aligning with the observed results from the device tests.

According to the suggestions, we have compared the power densities of reported thermoelectric flexible devices, as illustrated in **Table R1 (Page 27 in the revised Supporting Information)**. This table has been added to the supplementary information. The related information has been revised as: “Remarkably, a high open-circuit voltage of 6 mV, a substantial output power of 65 nW, and a competitive power density of 1.5 mW cm^{-2} can be simultaneously achieved at a ΔT of 20 K^{26,27}. The comparison results are provided in Supplementary Table 3.” **(Revised manuscript, Page 14)**

Table R1. A summary of the power density of thermoelectric thin film devices.

Device materials	Output power density (mW cm^{-2})
Cu ₃ Se ₁ single leg ²⁸	~0.09 ($\Delta T = 30 \text{ }^\circ\text{C}$)
N-Bi ₂ Te ₃ +P-Sb ₂ Te ₃ ²⁹	~0.1 ($\Delta T = 20 \text{ }^\circ\text{C}$)
N-Bi ₂ Te ₃ (PEDOT:PSS)+P-Sb ₂ Te ₃ (PEDOT:PSS) ³⁰	~0.2 ($\Delta T = 20 \text{ }^\circ\text{C}$)
Bi ₂ Te ₃ with 1wt% Se single leg ³¹	~0.14 ($\Delta T = 20 \text{ }^\circ\text{C}$)
P-Sb + N-Ni ³²	~0.9 ($\Delta T = 20 \text{ }^\circ\text{C}$)
N-Bi ₂ Te ₃ +P-Sb ₂ Te ₃ ³³	~1.42 ($\Delta T = 60 \text{ }^\circ\text{C}$)
Ag ₂ Se single leg ³⁴	~0.16 ($\Delta T = 20 \text{ }^\circ\text{C}$)
Ag ₂ Se single leg ²⁴	~0.23 ($\Delta T = 30 \text{ }^\circ\text{C}$)

Ag ₂ Se single leg ³⁵	~1.0 ($\Delta T = 20$ °C)
Ag ₂ Se single leg ³⁶	~1.4 ($\Delta T = 20$ °C)
This work Ag ₂ Se + Sb ₂ Te ₃	1.5 ($\Delta T = 20$ °C)

Comment 7: Why select Sb₂Te₃ as the other leg?

Author reply: Traditional thermoelectric devices typically require both n- and p-type legs³⁷⁻³⁹. To achieve higher performance in a specific temperature range, the temperature range corresponding to the highest performance of p- and n-type materials should be consistent or close. In this work, we prepared n-type Ag₂Se thin films with excellent performance near room temperature. According to our previous reports⁴⁰⁻⁴², the p-type Sb₂Te₃ film prepared in our laboratory exhibits optimal thermoelectric properties near room temperature, making it suitable as a p-type leg to match with Ag₂Se, given the similar temperature ranges of optimal thermoelectric performance.

Comment 8: The calculated surface energy of (00 \bar{l}) is reported as 0.002~0.004 eV/Å², which seems quite small? How about other materials?

Author reply: The surface energy was calculated by:

$$\gamma_s = \frac{1}{2A} (E_{slab} - NE_{atom}) \quad (14)$$

where A is the area of the surface. E_{slab} is the energy of the slab. N is the number of metal atoms. E_{atom} is the energy for one atom (Ag, Se, and Te) referred to the corresponding bulk structures. Based on the first-principles theory, the surface energy of (00 \bar{l}) is 0.004 eV/Å² (or 0.064 J/m²), and reduced to 0.002eV/Å² (or 0.032 J/m²) after Te doping, indicating the higher growth speed of (00 \bar{l}) planes after doping. In response to your comments, we were unable to find reports specifically addressing the

surface energy of Ag₂Se. However, we have compared results from other reported materials, as presented in **Table R2** (**Table 4** on **Page 28** in the revised Supporting Information). Our observations indicate that the surface of (00 \bar{l}) in Ag₂Se is small, suggesting a preference for the growth of this crystal plane. As for the reason for the calculated surface energy being relatively small, generally speaking, different calculation methods may lead to varied surface energy results, which is understandable. As long as the calculated variations follow a systematic pattern, even if the overall results are somewhat underestimated, the underlying mechanism can still be explained.

Table R2. A summary of reported surface energies of different materials.

Materials	Surface Energy (J m⁻²)
Mg ₂ Si (100) ⁴³	0.8-2.0
Al ₃ Ti (010) ⁴⁴	1.798
Al ₃ Ti (001) ⁴⁴	2.374/1.167
Boron Carbide Polytype B ₁₁ C _p (CBC) (10 $\bar{1}$ 1) ⁴⁵	3.21
Co (0001) ⁴⁶	2.110
Co ₃ Cr (0001) ⁴⁶	2.170
Co ₃ Mn (0001) ⁴⁶	2.766
Co ₃ Ni (0001) ⁴⁶	2.012
CoSb ₃ (100) ⁴⁷	0.330
CoSb ₃ (110) ⁴⁷	0.138
CoSb ₃ (111) ⁴⁷	0.797
Ni ₃ Nb (100) ⁴⁸	2.51
Ti ₂ AlC (0001) ⁴⁹	0.078

This work Ag₂Se (00l)

0.032-0.064

(0.002 eV/Å²-0.004 eV/Å²)

References:

1. Linseis, V., Völklein, F., Reith, H., Nielsch, K. & Woias, P. Advanced platform for the in-plane ZT measurement of thin films. *Rev. Sci. Instrum.* **89**, 015110 (2018).
2. Völklein, F., Reith, H. & Meier, A. Measuring methods for the investigation of in-plane and cross-plane thermal conductivity of thin films. *Phys. Status Solidi A* **210**, 106-118 (2013).
3. Völklein, F. Thermal conductivity and diffusivity of a thin film $\text{SiO}_2/\text{Si}_3\text{N}_4$ sandwich system. *Thin Solid Films* **188**, 27-33 (1990).
4. Crovetto, A. et al. Crystallize It before It Diffuses: Kinetic Stabilization of Thin-Film Phosphorus-Rich Semiconductor CuP_2 . *J. Am. Chem. Soc.* **144**, 13334-13343 (2022).
5. Yang, D. et al. High thermoelectric performance of aluminum-doped cuprous selenide thin films with exceptional flexibility for wearable applications. *Nano Energy* **117**, 108930 (2023).
6. Yang, J. et al. Low-Temperature ALD of $\text{SbO}_x/\text{Sb}_2\text{Te}_3$ Multilayers with Boosted Thermoelectric Performance. *Small*, 2306350 (2023). <https://doi.org/10.1002/sml.202306350>.
7. Yoon, S.E. et al. Remarkable Electrical Conductivity Increase and Pure Metallic Properties from Semiconducting Colloidal Nanocrystals by Cation Exchange for Solution-Processable Optoelectronic Applications. *Small* **19**, 2207511 (2023).
8. Zhang, X. et al. Waterborne Paints Based on Polymeric Semiconductor for Attachable Thermoelectric Generators. *Small Structures* **4**, 2200278 (2023).
9. Sikora, A. et al. Highly sensitive thermal conductivity measurements of suspended membranes (SiN and diamond) using a 3ω -Völklein method. *Rev. Sci. Instrum.* **83**, 054902 (2012).
10. Weathers, A. et al. Significant Electronic Thermal Transport in the Conducting Polymer Poly(3,4-ethylenedioxythiophene). *Adv. Mater.* **27**, 2101-2106 (2015).

11. Li, D. et al. Ce-filled $\text{Ni}_{1.5}\text{Co}_{2.5}\text{Sb}_{12}$ Skutterudite Thin Films with Record-High Figure of Merit And Device Performance. *Adv. Energy Mater.* **13**, 2301525 (2023).
12. Kitagawa, H. et al. Temperature dependence of thermoelectric properties of Ni-doped CoSb_3 . *J. Phys. Chem. Solids* **66**, 1635-1639 (2005).
13. Queisser, H.J.Haller, E.E. Defects in Semiconductors: Some Fatal, Some Vital. *Science* **281**, 945-950 (1998).
14. Ahtee, M., Nurmela, M., Suortti, P. & Jarvinen, M. Correction for preferred orientation in Rietveld refinement. *J. Appl. Crystallogr.* **22**, 261-268 (1989).
15. May, A.F.Snyder, G.J. Introduction to modeling thermoelectric transport at high temperatures. *Materials, preparation, and characterization in thermoelectrics* 1-18 (2012).
16. Lin, S. et al. Revealing the promising near-room-temperature thermoelectric performance in Ag_2Se single crystals. *J. Materiomics* **9**, 754-761 (2023).
17. Xie, Q.-Y. et al. Low thermal conductivity and high performance anisotropic thermoelectric properties of XSe ($\text{X} = \text{Cu}, \text{Ag}, \text{Au}$) monolayers. *Phys. Chem. Chem. Phys.* **24**, 7303-7310 (2022).
18. Jood, P., Chetty, R. & Ohta, M. Structural stability enables high thermoelectric performance in room temperature Ag_2Se . *J. Mater. Chem. A* **8**, 13024-13037 (2020).
19. Yu, K. et al. Ultra-low lattice thermal conductivity and enhanced thermoelectric performance in $\text{Ag}_{2-x}\text{Se}_{1/3}\text{S}_{1/3}\text{Te}_{1/3}$ via anion permutation and cation modulation. *J. Alloys Compd.* **885**, 161378 (2021).
20. Santhosh, R. et al. Controlled grain boundary interfaces of reduced graphene oxide in Ag_2Se matrix for low lattice thermal conductivity and enhanced power factor for thermoelectric

- applications. *J. Power Sources* **525**, 231045 (2022).
21. Santhosh, R. et al. Enhanced thermoelectric performance of hot-pressed n-type Ag₂Se nanostructures by controlling the intrinsic lattice defects. *CrystEngComm* **25**, 3317-3327 (2023).
 22. Day, T. et al. Evaluating the potential for high thermoelectric efficiency of silver selenide. *J. Mater. Chem. C* **1**, 7568-7573 (2013).
 23. Perez-Taborda, J.A., Caballero-Calero, O., Vera-Londono, L., Briones, F. & Martin-Gonzalez, M. High Thermoelectric zT in n-Type Silver Selenide films at Room Temperature. *Adv. Energy Mater.* **8**, 1702024 (2018).
 24. Ding, Y. et al. High Performance n-Type Ag₂Se Film on Nylon Membrane for Flexible Thermoelectric Power Generator. *Nat. Commun.* **10**, 841 (2019).
 25. Wei, T.-R., Qiu, P., Zhao, K., Shi, X. & Chen, L. Ag₂Q-Based (Q = S, Se, Te) Silver Chalcogenide Thermoelectric Materials. *Adv. Mater.* **35**, 2110236 (2023).
 26. Zhang, L., Shi, X.-L., Yang, Y.-L. & Chen, Z.-G. Flexible thermoelectric materials and devices: From materials to applications. *Mater. Today* **46**, 62-108 (2021).
 27. Cao, T., Shi, X.-L. & Chen, Z.-G. Advances in the design and assembly of flexible thermoelectric device. *Prog. Mater. Sci.* **131**, 101003 (2023).
 28. Lu, Y. et al. Good Performance and Flexible PEDOT:PSS/Cu₂Se Nanowire Thermoelectric Composite Films. *ACS Appl. Mater. Interfaces* **11**, 12819-12829 (2019).
 29. Yu, Y. et al. Towards high integration and power density: Zigzag-type thin-film thermoelectric generator assisted by rapid pulse laser patterning technique. *Appl. Energ.* **275**, 115404 (2020).
 30. We, J.H., Kim, S.J. & Cho, B.J. Hybrid Composite of Screen-Printed Inorganic Thermoelectric

- Film and Organic Conducting Polymer for Flexible Thermoelectric Power Generator. *Energy* **73**, 506-512 (2014).
31. Madan, D. et al. Enhanced Performance of Dispenser Printed MA n-type Bi₂Te₃ Composite Thermoelectric Generators. *ACS Appl. Mater. Interfaces* **4**, 6117-6124 (2012).
 32. Cao, J. et al. Flexible elemental thermoelectrics with ultra-high power density. *Mater. Today Energy* **25**, 100964 (2022).
 33. Ao, D.-W. et al. Assembly-Free Fabrication of High-Performance Flexible Inorganic Thin-Film Thermoelectric Device Prepared by a Thermal Diffusion. *Adv. Energy Mater.* **12**, 2202731 (2022).
 34. Lu, Y. et al. Ultrahigh power factor and flexible silver selenide-based composite film for thermoelectric devices. *Energy Environ. Sci.* **13**, 1240-1249 (2020).
 35. Lei, Y. et al. Microstructurally Tailored Thin β -Ag₂Se Films towards Commercial Flexible Thermoelectrics. *Adv. Mater.* **34**, 2104786 (2021).
 36. Hou, S. et al. High-performance, thin-film thermoelectric generator with self-healing ability for body-heat harvesting. *Cell Rep. Phys. Sci.* **3**, 101146 (2022).
 37. He, R., Schierning, G. & Nielsch, K. Thermoelectric Devices: A Review of Devices, Architectures, and Contact Optimization. *Adv. Mater. Technol.* **3**, 1700256 (2018).
 38. Kishore, R.A., Nozariasbmarz, A., Poudel, B., Sanghadasa, M. & Priya, S. Ultra-high performance wearable thermoelectric coolers with less materials. *Nat. Commun.* **10**, 1765 (2019).
 39. Rogl, G. et al. Changes in microstructure and physical properties of skutterudites after severe plastic deformation. *Phys. Chem. Chem. Phys.* **17**, 3715-3722 (2015).

40. Wei, M. et al. Directional Thermal Diffusion Realizing Inorganic Sb₂Te₃/Te Hybrid Thin Films with High Thermoelectric Performance and Flexibility. *Adv. Funct. Mater.* **32**, 2207903 (2022).
41. Li, Y.-l. et al. Optimized thermoelectric properties of flexible p-type Sb₂Te₃ thin film prepared by a facile thermal diffusion method. *J. Alloys Compd.* **948**, 169730 (2023).
42. Zheng, Z.-h., Luo, J.-t., Li, F., Liang, G.-x. & Fan, P. Enhanced thermoelectric performance of P-type Sb₂Te₃ thin films through organic-inorganic hybridization on flexible substrate. *Curr. Appl. Phys.* **19**, 470-474 (2019).
43. Liao, J.N., Li, K., Wang, F., Zeng, X.S. & Zhou, N.G. Properties of Mg₂Si (100) surfaces: A first-principles study. *Solid State Commun.* **183**, 41-46 (2014).
44. Qin, J. Partition the total energy of Al₃Ti to characterize the Al₃Ti/Al interface properties: a first-principles study. *Surf. Sci.* **739**, 122398 (2024).
45. Beaudet, T.D., Smith, J.R. & Adams, J.W. Surface energy and relaxation in boron carbide (10 $\bar{1}$) from first principles. *Solid State Commun.* **219**, 43-47 (2015).
46. Cao, Y., Lin, Q., Huang, Q., Xu, Y. & Zhou, S. First-principles predictions of corrosion resistance of (0001) surface of Co and Co₃X (X = Cr, Ni, Mn) compounds. *Computational and Theoretical Chemistry* **1225**, 114171 (2023).
47. Hammerschmidt, L., Quennet, M., Töpfer, K. & Paulus, B. Low-index surfaces of CoSb₃ skutterudites from first principles. *Surf. Sci.* **637-638**, 124-131 (2015).
48. Hao, L., Chen, W., Lei, X., Yao, W. & Wang, N. Structures, Energies, and Electronic Properties of Low-Index Surfaces of γ ''-Ni₃Nb: A First-Principles Calculations. *physica status solidi (b)* **260**, 2300239 (2023).
49. Liu, P. et al. A systematic investigation on the surface properties of Ti₂AlC *via* first-principles

calculations. *Surf. Sci.* **735**, 122337 (2023).

REVIEWERS' COMMENTS

Reviewer #1 (Remarks to the Author):

The manuscript is well revised. I recommend the publication of this manuscript in Nature Communications.

Reviewer #2 (Remarks to the Author):

The authors have addressed all my comments. Thus, I recommend accepting this article for publication.

Reviewer #3 (Remarks to the Author):

I think the authors have responded well to the comments of the reviewers. I'd like to recommend the publication of this paper.